# Generalised optical printing of photocurable metal chalcogenides

Seongheon Baek[1,5], Hyeong Woo Ban[1,5], Sanggyun Jeong[2,5], Seung Hwae Heo[1], Da Hwi Gu[1], Wooyong Choi[1], Seungjun Choo[1], Yae Eun Park[3], Jisu Yoo[1], Moon Kee Choi[1,3,4], Jiseok Lee [2] ✉ & Jae Sung Son [1,3] ✉

Optical three-dimensional (3D) printing techniques have attracted tremendous attention owing to their applicability to mask-less additive manufacturing, which enables the cost-effective and straightforward creation of patterned architectures. However, despite their potential use as alternatives to traditional lithography, the printable materials obtained from these methods are strictly limited to photocurable resins, thereby restricting the functionality of the printed objects and their application areas. Herein, we report a generalised direct optical printing technique to obtain functional metal chalcogenides via digital light processing. We developed universally applicable photocurable chalcogenidometallate inks that could be directly used to create 2D patterns or micrometre-thick 2.5D architectures of various sizes and shapes. Our process is applicable to a diverse range of functional metal chalcogenides for compound semiconductors and 2D transition-metal dichalcogenides. We then demonstrated the feasibility of our technique by fabricating and evaluating a micro-scale thermoelectric generator bearing tens of patterned semiconductors. Our approach shows potential for simple and cost-effective architecturing of functional inorganic materials.

Material patterning is considered an essential prerequisite for fabricating electronic, optoelectronic, and energy devices[1–5]. Various methods, such as photolithography, imprint lithography, microcontact printing, and laser or electron-beam writing, have been developed to create high-resolution patterned structures in two-dimensional (2D) films[6–11]. However, these conventional manufacturing processes, which are based on mask or mould production, lithography, vacuum deposition, and lift-off, are often costly and require multi-step processes or expensive lithography equipment[11,12]. Furthermore, these 2D design processes are unsuitable for fabricating three-dimensional (3D) structures, which usually involve high costs and long processing times, when performed by traditional subtractive lithography. Optical 3D printing techniques, such as

digital light processing (DLP) and stereolithography apparatus (SLA), have recently been regarded as simple and cost-effective patterning methods by additive manufacturing[13,14]. In these methods, automatically patterned digital masks produced by a simple electrical input create solid architectures directly from liquid-type resin inks following light exposure; thus, mask production, material deposition, and subsequent lift-off are unnecessary. Despite these advantages, however, optical printing processes suffer from the critical issue of limited printable materials. Only photocurable polymers or their composites with inorganic fillers are utilised as inks to ensure photocurability[15–17]. Thus, the use of these processes to create functional and high-performance inorganic material patterns is limited, and their applications in the electronic and energy fields are

[1]Department of Materials Science and Engineering, Ulsan National Institute of Science and Technology (UNIST), Ulsan 44919, Republic of Korea. [2]School of Energy and Chemical Engineering, Ulsan National Institute of Science and Technology (UNIST), Ulsan, Republic of Korea. [3]Graduate School of Semiconductor Materials and Devices, Ulsan National Institute of Science and Technology (UNIST), Ulsan 44919, Republic of Korea. [4]Center for Nanoparticle Research, Institute for Basic Science (IBS), Seoul 08826, Republic of Korea. [5]These authors contributed equally: Seongheon Baek, Hyeong Woo Ban, Sanggyun Jeong. ✉e-mail: jiseok@unist.ac.kr; jsson@unist.ac.kr

restricted. This is in stark contrast to the direct ink writing (DIW) techniques, which is an extrusion-based 3D printing method to create meso- and micro-scales architectures. By developing the viscoelastic inks containing inorganic particles, the DIW has been extensively utilized for fabricating various types of functional inorganic materials[18–20]. For example, the DLP-printed thermoelectric BiSbTe materials was reported to exhibit the *ZT* value of 0.12, which is one order of magnitude lower than that of inorganic bulk reference materials or that of even DIW-printed BiSbTe because the printed objects contain organic residues[21]. Calcination in air or oxygen can completely remove organic residues, but this process can also cause the undesirable oxidation of inorganic materials, such as metal chalcogenides. Nevertheless, the optical printing methods of DLP and SLA can be advantageous for high-resolution, high-throughput, and large-scale printing, which can offer great potential for patterning high-performance inorganic materials.

Metal chalcogenides have attracted significant interest in various disciplines as well-defined semiconductor materials that are useful in various technological applications, such as electronics, optoelectronics, and thermoelectrics[22–26]. Tremendous efforts have been made to pattern metal chalcogenides using various lithographic and printing techniques[27–31]. For example, Wang et al. reported the direct optical lithography of a functional inorganic nanomaterial; in this work, spin-coated photocurable nanoparticle thin films were easily patterned by photolithography and subsequent lift-off[27]. Similarly, Yang et al. developed a high-resolution optical patterning process for photocurable CdSe-based quantum dots mixed with an azide-based light-driven ligand crosslinker; in these authors' work, the dots were patterned through photolithography and subsequent lift-off[28]. Although these methods can provide high-resolution patterns in 2D inorganic films, they still adopt conventional multi-step processes, which present challenges within the context of process simplification and 3D printability.

Chalcogenidometallates (ChaMs) have been intensively studied as soluble semiconductor inorganics that could provide metal chalcogenide inks for various printing processes. In general, these molecular anions are composed of metallic atom centres coordinated with several chalcogens and readily form crystalline metal chalcogenide phases via thermal decomposition[32–36]. These unique characteristics of ChaMs render them suitable candidates as precursors for the ink-processed patterning of metal chalcogenides. Herein, we report a mask-less DLP-based printing technology to obtain organics-free inorganic metal chalcogenides with 2D films and 2.5D architectures. We developed photocurable ChaM-based inorganic inks with the aid of a photoacid generator (PAG), which enables the fabrication of complex 2D-patterned metal chalcogenides with excellent fidelity and size and shape uniformity as well as micrometre-thick 2.5D architectures by 2D layer-by-layer additive manufacturing. This process was demonstrated to be universally applicable to the diverse metal chalcogenides of functional compound semiconductors and 2D layered transition metal dichalcogenides. To verify the feasibility of our proposed approach, we fabricated in-plane and cross-plane micro-scale thermoelectric power generator composed of patterned SnSe₂ and Cu₂S legs as n-type and p-type semiconductors, respectively, by DLP printing. This generator exhibited a power density of 0.564 mW cm⁻² under a temperature difference of 65 K. Our approach shows great potential use as a cost-effective, simple, and high-resolution direct 2D and 3D architecturing process for inorganic materials.

## Results

### Synthesis of photocurable chalcogenidometallate-based inks

The overall process of the proposed optical printing technique using photocurable ChaM-based inks is illustrated in Fig. 1a. We used a DLP process that operates by reflecting light off microscopic mirrored panels called digital micro-mirror devices to selectively solidify the photocurable ink and create complex 3D architectures by 2D layer-by-layer curing. We synthesised diverse ranges of ChaMs containing Pt, Sb, Sn, Cu, and Mo by dissolving the corresponding metal chalcogenide powders using an ethylenediamine/ethanethiol alkahest in a $N_2$-filled glove box or traditional coordination chemistry to generate the desired ChaM anions counterbalanced by cations of ammonium or ethylendiammonium (Fig. 1a)[37,38]. All ChaMs exhibited excitonic peaks in the visible range of the UV-Vis absorption spectrum due to the d−d transition or ligand-to-metal charge transfer of the metal chalcogenide complexes (Supplementary Fig. 1)[39–41]. The characteristic peak at 520 nm observed in the spectra of the Sb-, Cu-, and Sn-based ChaMs originated from the alkahest solvent[42].

The synthesised ChaMs were mixed with an appropriate amount of PAGs to formulate the photocurable ChaM-based inks. Here, the purification step of the ChaMs is essential to ensure their photocurability in the presence of PAGs because unreacted metal chalcogenides or by-products could negatively affect the kinetics of the photoreaction of the ChaMs. In addition, the basic reaction medium of ethylenediamine or ammonia solution was replaced with typical polar solvents, such as *N*-methyl formamide (NMF) or dimethyl sulfoxide (DMSO), to ensure the photoreactivity of the PAGs. Solvent exchange was confirmed by UV-Vis absorption analysis, which showed the disappearance of the characteristic peak at 520 nm after complete purification (Supplementary Fig. 1)[42]. We used a typical nonionic triazine-based PAG, 2-[2-(5-methyl furan-2-yl) vinyl]−4,6-bis(trichloromethyl)−1,3,5-triazine (MFVT), and a sulfonate-based PAG, *N*-(trifluoromethylsulfonyloxy)−1,8-naphthalimide (IM-NIT), to prepare the inks. MFVT and IM-NIT have a wide absorption spectrum range of ~350–400 nm and, thus, are suitable for h-line (405 nm) or i-line (365 nm) light sources (Supplementary Fig. 2). The mixed inks maintained their colloidal stability for over 14 d under dark conditions (Supplementary Fig. 3). The UV-Vis spectra of the inks showed the peaks of both ChaMs and MFVT, which indicates that these components do not react without UV irradiation (Supplementary Fig. 4). Upon UV irradiation, MFVT decomposed to produce Cl−C • and Cl• radicals, which reacted with alcoholic solvent (ethanol, methanol, isopropanol, etc.) and non-alcoholic solvent (DMSO, acetonitrile, etc.) and released protons (H⁺) in the ink medium[43,44]. In the case of IM-NIT, heterolytic cleavage, rather than homolytic cleavage, of the N−O bond occurred owing to the presence of the triflate anion, which is a good leaving group. The ionic intermediates produced by this process generated a photoacid (Supplementary Fig. 5)[45,46]. Because chalcogens with the form $M_xCh_y^{n-}$ (Ch = S, Se) in the ChaMs have relatively high proton affinity (S: 1443–1453 kJ/mol, Se: 1408–1420 kJ/mol), the protons generated by the photoreaction of the PAGs rapidly react with the ChaM anions, causing precipitation of the ChaMs (Fig. 1b). The ξ-potential of the MoS₂-based inks decreased from −28.7 to −13.6 mV upon UV irradiation (Fig. 1c), which suggested that the ChaMs lost their charges following their reaction with protons. Moreover, the solute sizes measured by dynamic light scattering (DLS) significantly increased from 106 to 295 nm upon UV irradiation (Fig. 1d), thereby demonstrating the photoreaction-induced precipitation of the ChaMs (Fig. 1e).

### Optical printing of metal chalcogenides

Based on the photoreaction of the formulated ChaM-based inks, we applied the DLP method using our inks to achieve the optical patterning of metal chalcogenides. The digital masks designed for DLP printing in this study are shown in Supplementary Fig. 6. The scanning electron microscopy (SEM) images in Fig. 2a, b show that the widths of the printed line patterns of the PtS₂-based ChaMs range from 25 to 100 μm; moreover, the minimum width observed is close to the equipment limit of 10 μm. Various printed patterns of circles, squares, and triangles ranging in size from tens to hundreds of micrometres were also obtained. All patterned structures exhibited excellent shape fidelity and uniformity. We fabricated a large square pattern with

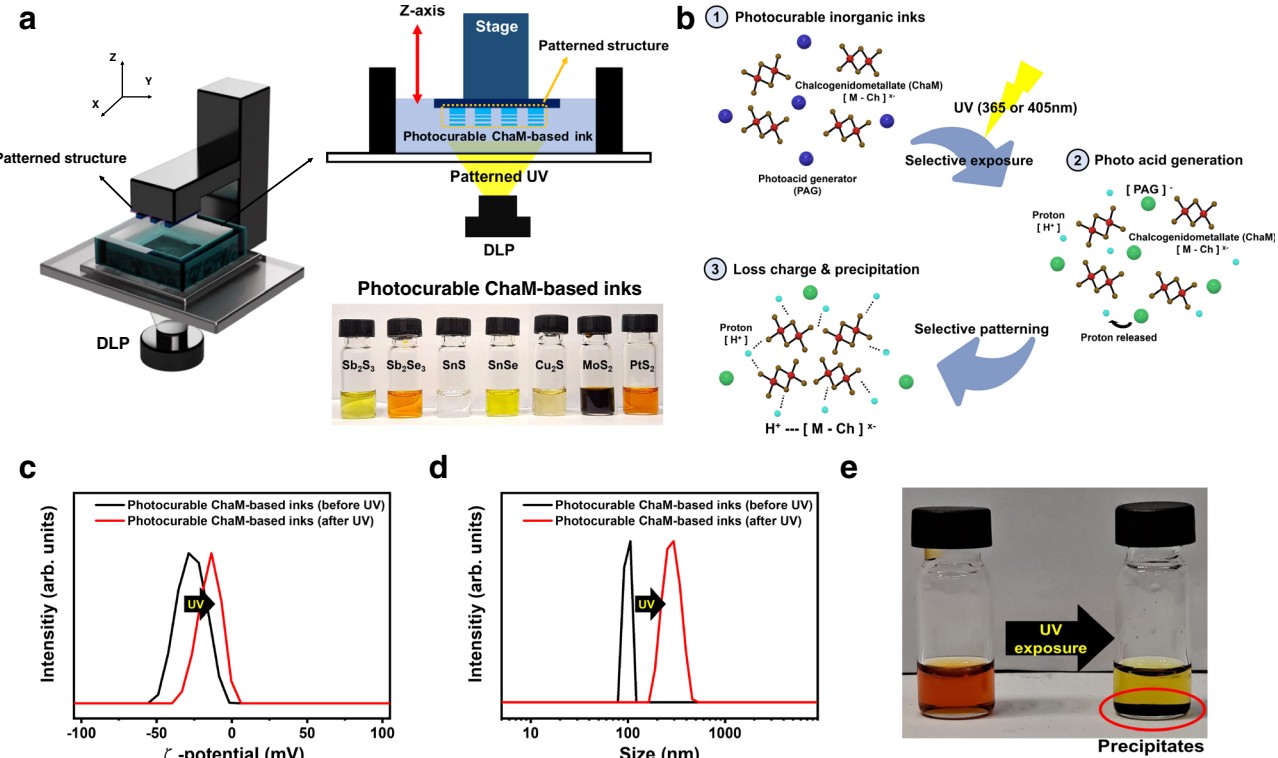

**Fig. 1 | Generalised optical printing of metal chalcogenide. a** Schematic of the digital light processing (DLP)-based optical printing of chalcogenidometallate (ChaM)-based inks. The DLP process that operates by reflecting light off digital micro-mirror devices selectively solidifies the photocurable metal chalcogenide inks and create complex 2D films and 2.5D architectures by 2D layer-by-layer curing. **b** Photocuring mechanism of the photocurable ChaM-based inks. The ChaMs react with the protons generated by the photoreaction of the photoacid generators (PAGs), causing precipitation of the ChaMs. **c** ξ-potential (zeta potential), **d** Dynamic light scattering (DLS) size, and **e** photograph of the photocurable ChaM-based inks before (black line) and after (red line) ultraviolet (UV) irradiation.

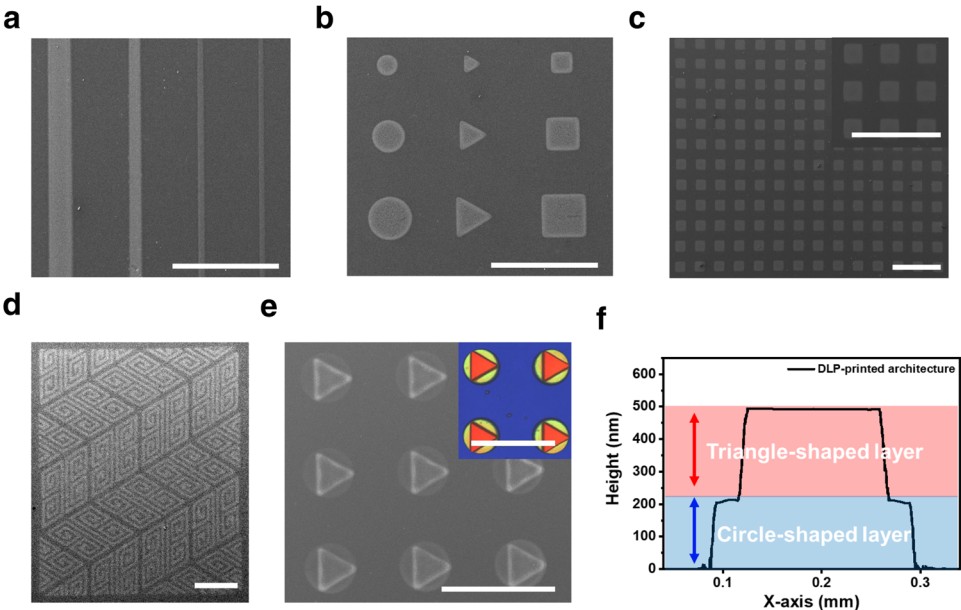

**Fig. 2 | DLP printing of ChaM-based inks. a** Scanning electron microscopy (SEM) image of the printed line patterns of PtS₂-based chalcogenidometallte (ChaM) ink with widths ranging from 100 μm to 25 μm. **b** SEM image of the printed circles, squares, and triangles of PtS₂-based ChaM ink at different scales of tens to hundreds of micrometres. **c** SEM image of hundreds of printed square array patterns with a width of 100 μm at a scale of several millimetres. **d** SEM image of the printed geometric pattern of MoS₂-based ChaM ink with a linewidth of 50 μm created by the digital light processing (DLP) method. **e** SEM and 3D scan analysis (inset) images of the printed 2.5D architecture composed of circle- and triangle-shaped layers. **f** Height profile of the printed 2.5D architecture. All scale bars in the a–e indicate 500 μm.

dimensions of 1.2 cm × 1.2 cm. The root-mean-square surface roughness of the entire printed structure was as low as 4.66 nm, which demonstrated the excellent uniformity of the proposed printing process over a centimetre-scale area (Supplementary Fig. 7). Another important advantage of our optical printing process is the high-throughput fabrication of multiple patterns on a single substrate. We successfully realised the printing of hundreds of 100 μm-wide square arrays on a substrate in several millimetres area (Fig. 2c). These examples of high-quality 2D patterns confirm that our cost-effective and simple patterning method for 2D films may be an alternative approach to traditional lithographic techniques. As the minimum linewidths and maximum sizes of the DLP-printed patterns critically depend on the equipment specifications, we speculate that higher-resolution and larger-scale equipment could create finer and larger-scale patterns with our inks.

The direct printability of the ChaM-based inks allows for the printing of complex 2D patterns with multiple materials and 2.5D architectures via layer-by-layer curing. As shown in the SEM image in Fig. 2d, a geometric pattern of the $MoS_2$-based ChaM with a linewidth of 50 μm was created using the DLP method. Interferometric scattering-based 3D scanning analysis of the geometric pattern obtained revealed a thickness of 45 nm with high lateral fidelity (Supplementary Fig. 8). Multiple material patterns were also successfully created on a single substrate. For example, a maze and its solution line were patterned using different ChaMs via the sequential DLP printing of $MoS_2$- and $PtS_2$-based ChaM inks. Specifically, we first patterned a $MoS_2$-based ChaM maze with a linewidth of 85 μm and then patterned the solution line with a thinner linewidth of 55 μm over this maze using $PtS_2$-based ChaM ink (Supplementary Fig. 9). Moreover, we demonstrated the compatibility of our approach to 3D printing by layer-by-layer sequential deposition of ~50 nm-thick layers on the substrate. Multiple 2.5D architectures composed of triangular layers printed over circular layers with a thickness of 500 nm were printed on a single substrate. The SEM and 3D scanning images in Fig. 2e show the excellent shape uniformity and fidelity of all architectures obtained. The 3D scan height profile in Fig. 2f also revealed the lateral shape fidelity of the printed architectures, which suggests minimal distortion of the pre-printed layers during the subsequent deposition process. In addition, the micrometre-scale architectures were fabricated by the layer-by-layer DLP printing. As shown in the SEM images and the height profile (Supplementary Fig. 10), we printed the 1.5 μm-thick pyramidal 3D architecture constructed by thirty layers of $MoS_2$-based ChaMs. These pyramids consist of three squares with different lengths of 500, 400, and 300 μm. These results demonstrate the compatibility of our approach to 3D printing of inorganic metal chalcogenides.

## Material characterisations of the printed metal chalcogenides

Given to the universal applicability of the photocuring mechanism, we printed a variety of ChaMs using our DLP method. The thermal decomposition characteristics of the ChaMs, crystalline metal chalcogenides, such as the functional compound semiconductors of $Sb_2S_3$, $Sb_2Se_3$, $Cu_2S$, SnS, and SnSe, and 2D transition-metal dichalcogenides, including $SnSe_2$, $MoS_2$, and $PtS_2$, were obtained by subjecting the patterned ChaMs to post-heating treatment under an inert atmosphere at temperatures of 473–623 K[47]. The characters of the chemical formulas corresponding to the studied materials were printed, as shown in the insets of Fig. 3a–f. Optical microscopy (OM) analysis revealed the absence of undesirable microstructural evolution in the printed structures during heat treatment (Supplementary Fig. 11). Moreover, the composition of the metal chalcogenides could be modulated by controlling the conditions of the heat treatment. Specifically, varying the temperature or holding time allowed us to control the degree of evaporation of the chalcogen and, thus, its content in the final product. For example, heat treatment of the same sample at 573 and 673 K produced the $SnSe_2$ and SnSe phases, respectively.

The high-magnification SEM images of all samples revealed smooth and dense microstructures without the substantial formation of voids and pores (Fig. 3a–h). In particular, the printed SnSe, $SnSe_2$, and SnS films showed a 2D plate-like morphology, which can be attributed to the active lateral 2D growth of the grains, which are well known to have 2D layered structures. This unique microstructure was supported by the X-ray diffraction (XRD) patterns of the samples with the thickness of 250 nm (Supplementary Fig. 12), which showed peaks corresponding to specific axes. For example, the XRD patterns of SnSe (Fig. 3c) showed the c-axis peaks of the (200), (400), and (800) planes, indicating strong texturing in the microstructures of these samples. Also, the $SnSe_2$ exhibited the a-axis peaks of the (001), (003), and (004) planes in the XRD pattern (Fig. 3d), showing a strong orientation when it compared with that of the non-textured $SnSe_2$ phase (Supplementary Fig. 13).

In addition, the XRD patterns of the other samples exhibited peaks corresponding to those of the bulk references, which indicates their high crystallinity. The 2D transition-metal dichalcogenides $MoS_2$ and $PtS_2$ were characterised by Raman spectroscopy. The Raman spectrum of the printed $MoS_2$ layer showed peaks at ~380 and ~403 $cm^{-1}$, which correspond to the $E_{2g}$ and $A_{1g}$ vibration modes of $2H-MoS_2$ (Fig. 3g)[48]. The two intense peaks at ~300 and ~339 $cm^{-1}$ in the Raman spectrum of the $PtS_2$ thin film were well matched to the $E_{1g}$ and $A_{1g}$ modes, respectively, of the $PtS_2$ reference phase (Fig. 3h)[49]. These results demonstrate the high crystallinity and controllable micro-scale crystallographic textures of the patterned metal dichalcogenides obtained via our process.

## Functional properties of the printed metal chalcogenides

The functionality of the patterned metal chalcogenide semiconductors was demonstrated in terms of their electrical properties, which were obtained using Hall measurements; the charge carrier mobility and concentration of these materials were then obtained. We selected $Cu_2S$ and $SnSe_2$ as model p-type and n-type semiconductors, respectively, because they are known to be low-cost semiconductors exhibiting good electrical properties[50,51]. Here, we fabricated 2D $Cu_2S$ and $SnSe_2$ samples using the DLP method and then annealed $Cu_2S$ at 723 for 5, 10, and 15 min, and $SnSe_2$ at 573 K for 7, 10, and 15 min, respectively. The annealing temperatures were selected after considering the thermal stability of the materials to conserve their stoichiometric composition and microstructural integrity. The annealing conditions are further discussed in the Supplementary Discussion and in Supplementary Fig. 14 and 15. The Hall measurements of the printed $Cu_2S$ clearly indicated p-type characteristics. As the annealing time increased from 5 to 15 min, the room-temperature hole concentration of the samples decreased from $1.58 \times 10^{20}$ to $6.29 \times 10^{19}$ $cm^{-3}$ (Fig. 4a), which agrees with the reported hole concentration range of bulk $Cu_2S$[52]. Because the intrinsic defects of Cu vacancies in $Cu_2S$ act as hole donors, longer annealing times could promote the evaporation of S, eventually decreasing the hole concentrations by increasing the relative Cu contents and reducing the Cu vacancy defects[53]. The hole mobility of the samples also increased from 1.29 to 3.89 $cm^2$ $V^{-1}$ $s^{-1}$ with increasing annealing time (Fig. 4a), likely because longer annealing times could decrease the number of Cu vacancy point defects and lead to higher crystallinity. Moreover, the SEM images of the samples (Supplementary Fig. 16) showed fewer pinholes and cracks in the microstructures of samples heated for longer durations, which further contributed to the increase of the carrier mobility. By comparison, the $SnSe_2$ samples showed n-type characteristics, and their electron concentration, at ~$1.0 \times 10^{19}$ $cm^{-3}$, remained nearly constant regardless of the annealing time (Fig. 4b). Furthermore, the electron mobility of the samples increased from 1.06 to 2.25 $cm^2$ $V^{-1}$ $s^{-1}$ as the annealing time increased (Fig. 4b), which suggests grain growth, as supported by their SEM images (Supplementary Fig. 17). In general, the maximum electron mobilities

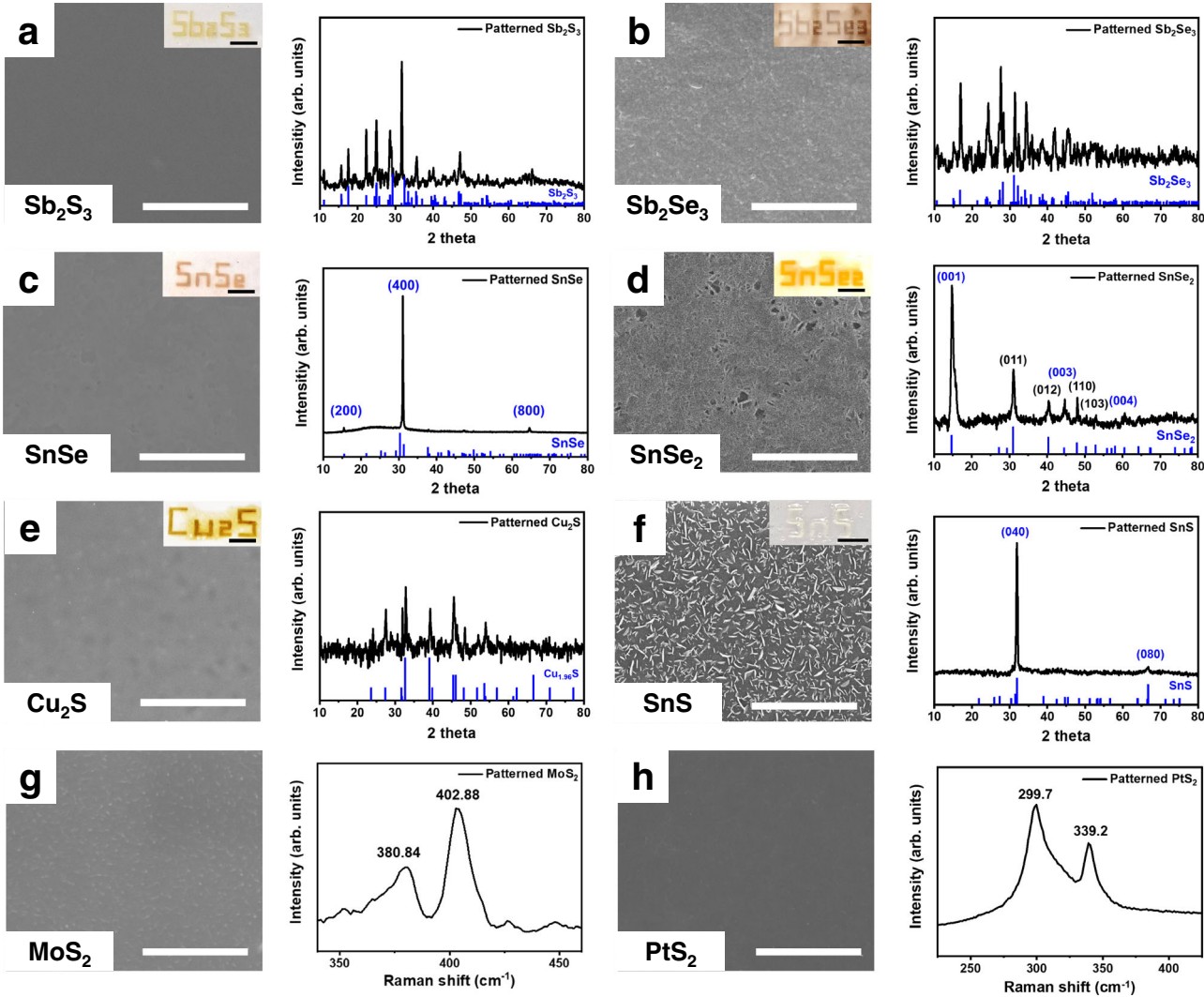

**Fig. 3 | Microstructures and crystallinity of the printed metal chalcogenides.** Scanning electron microscopy (SEM) images (left), X-ray diffraction (XRD) patterns (right), and photographs (inset) of the printed **a** Sb$_2$S$_3$, **b** Sb$_2$Se$_3$, **c** SnSe, **d** SnSe$_2$, **e** Cu$_2$S, and **f** SnS. The blue labels of the XRD patterns of panel **c**, **d**, and **f** indicate the peaks related to the crystalline orientation along c-axis of SnSe, a-axis of SnSe$_2$, and c-axis of SnS, respectively. The black labels in the XRD pattern of panel **d** indicate the peaks other than the a-axis. Vertical blue lines indicate the patterns for the corresponding bulk references (Sb$_2$S$_3$: JCPDS 03-065-2434, Sb$_2$Se$_3$: JCPDS 01-083-7430, SnSe: JCPDS 00-032-1382, SnSe$_2$: JCPDS 01-089-2939, Cu$_{1.96}$S: JCPDS 00-012-0224, SnS: JCPDS 00-014-0620). SEM images (left) and Raman spectra (right) of **g** MoS$_2$ and **h** PtS$_2$. (Scale bars: SEM images: 3 μm, inset images: 5 mm) The labels in panels **g** and **h** indicate the Raman shift of the peaks in the spectrum.

of the printed n- and p-type samples exceeded 1 cm$^2$ V$^{-1}$ s$^{-1}$ under optimal conditions.

The carrier mobilities of our printed samples are comparable with those reported for metal chalcogenide and inorganic patterns fabricated by traditional lithographic techniques or other printing methods, as summarised in Supplementary Table 1, and are comparable with or even higher than those of the reported solution-processes thin films or the 3D-printed samples with the same materials (Supplementary Table 2). Although the electrical properties such as mobility and conductivity of our samples were roughly an order of magnitude lower than the bulk values, this can be understood with consideration that the ink-processed materials are usually more defective and less dense and include smaller grains compared with bulk materials synthesized by energy-intensive methods under harsh condition. For example, the grain sizes of our Cu$_2$S and SnSe$_2$ samples range from several tens to several hundreds of nanometres, which are several orders of magnitudes lower than those observed in the reported bulks[52,54,55]. These smaller grains should cause the higher density of interfaces that can hinder the charge carrier transport. Moreover, various types of defects can be formed in these materials since the inks are the mixtures of

organic solvents and precursors. In our samples, the precursors are organometallic complexes of ChaMs with counter cation of ethylenediammonium. Thus, there can be undesired impurities residues in the fabricated samples, which can reduce the charge carrier mobility and electrical conductivity by the impurity scattering. Finally, the ink-processed materials generally have porous features in the microstructures; e.g. the SEM images of our samples shows multiple pin-holes in the nanometre scale. The formation of these pores is usually inevitable since they are created during the solvent drying or precursor decomposition that releases gases. All these reasons can be responsible to the reduced charge carrier mobility and resulting electrical conductivity.

Given the excellent electrical properties of the Cu$_2$S and SnSe$_2$ samples, they may be applied to energy devices, such as thermoelectric devices. These devices are of tremendous interest because they can achieve direct heat–electricity conversion, which is useful in various technological applications, such as energy harvesting, local thermal management, and thermal sensors[56–59]. The electronic devices that have emerged in recent years require autonomous energy systems, which, in turn, demand system-adaptable energy sources

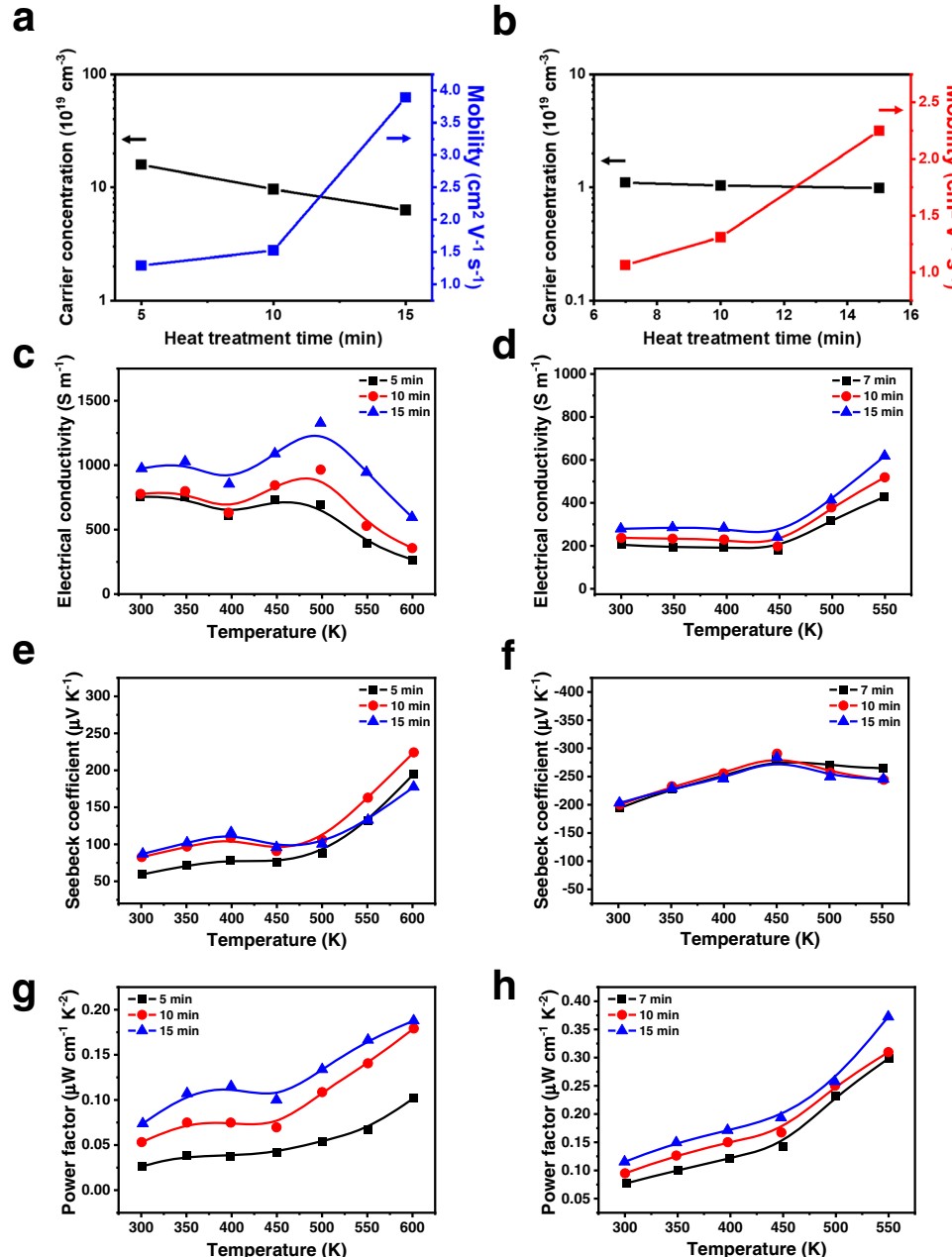

**Fig. 4 | Functionality of the printed metal chalcogenide semiconductors.** Charge carrier mobility and concentration of the printed **a** Cu₂S and **b** SnSe₂ as functions of the heat treatment time as detected by Hall effect measurements. Temperature dependences of **c**, **d** electrical conductivity, **e**, **f** Seebeck coefficient, and **g**, **h** power factor of the printed Cu₂S (c,e,g) and SnSe₂ (d,f,h) as functions of the heat treatment time.

integrated into them[60]. In this context, patterned micro thermoelectric generators may be ideal candidates as auxiliary power sources for these systems. The materials used in these devices must have high microstructural quality and tight engineering to preserve their thermoelectric properties and optimise their trade-off properties. Such materials would provide an excellent model system for demonstrating the feasibility of our process. Thus, we characterised the temperature-dependent thermoelectric properties of the $Cu_2S$ and $SnSe_2$ samples. The temperature-dependent electrical conductivity and Seebeck coefficients of the printed $Cu_2S$ samples were measured from room temperature to 600 K (Fig. 4c, e). As expected, the $Cu_2S$ samples exhibited p-type semiconductor behaviour, as indicated by their positive Seebeck coefficients. Moreover, fluctuations in both electrical conductivity and Seebeck coefficient were clearly observed in the temperature range of 350–500 K, in agreement with the typical

behaviour of $Cu_{2-x}S$ crystals, which exhibit a phase transition from the low chalcocite phase to the high chalcocite phase in the corresponding temperature range[51,52]. This phase transition reflects the good crystallinity of our $Cu_2S$ samples fabricated by DLP printing. Among the samples investigated, that annealed for 15 min showed the highest room-temperature electrical conductivity (1000 S m⁻¹) owing to its high carrier mobility and concentration (Fig. 4c). This value corresponds to ~20% of the electrical conductivity previously reported for $Cu_2S$ bulk samples prepared by spark plasma sintering[52]. The Seebeck coefficients of all samples showed a positive dependence on temperature owing to their high carrier concentration. Generally, the contribution of minority carriers to the Seebeck coefficient is negative and become more significant at higher temperatures by the thermal excitation, eventually causing the negative temperature dependence of the Seebeck coefficients. However, in the material with a high

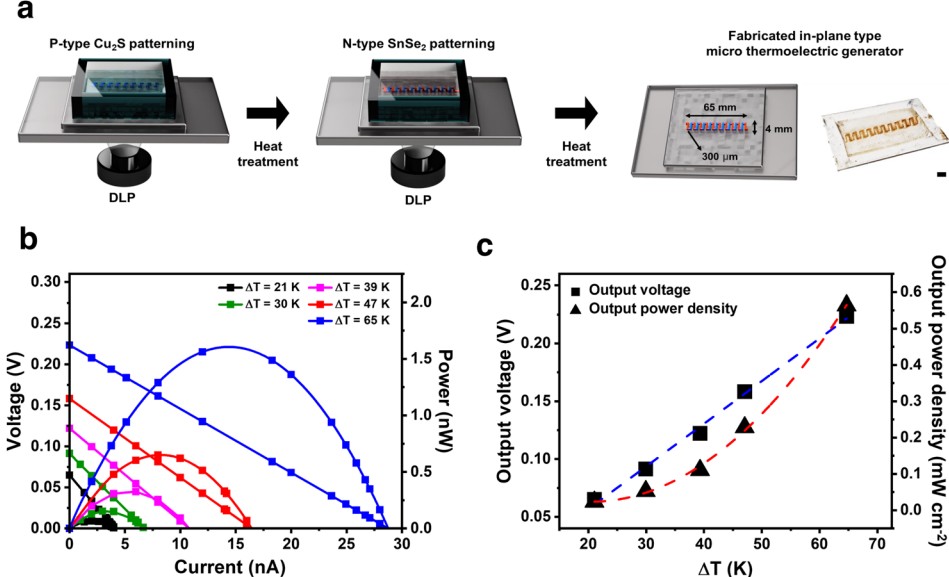

**Fig. 5 | Fabrication and evaluation of the micro-scale thermoelectric generator. a** Scheme of the fabrication of the micro-scale thermoelectric generator by DLP-based optical printing. (Scale bar: 5 mm) To fabricate the thermoelectric generator, the p-type $Cu_2S$ legs were initially patterned and heat-treated at 723 K, and then the n-type $SnSe_2$ legs were patterned and annealed at 573 K. **b** Output voltages and powers of the micro-scale thermoelectric generator at various temperature differences. **c** Plots of output voltage and power density as functions of temperature difference. The blue dashed, and red dashed lines were fitted by a linear fitting of the output voltage and a second-order polynomial fitting of output power, respectively.

concentration of the majority carriers, the minority carriers affect less the electrical transport by suppressing the bipolar effect. Accordingly, these materials usually show the positive temperature dependences in a wide temperature range. This behaviour is generally reflected in the temperature-dependent Seebeck coefficients of materials, which shows the shift of the peak value of the Seebeck coefficients to higher temperature ranges with the increase of carrier concentrations of materials[61,62]. The sample annealed for 10 min showed a maximum Seebeck coefficient of 230 μV $K^{-1}$ at 600 K (Fig. 4e), which is comparable with the reported value for $Cu_2S$ bulk materials[52]. The sample annealed for 15 min also showed the highest power factor (0.187 μW $cm^{-1}$ $K^{-2}$) at 600 K (Fig. 4g).

The temperature-dependent electrical conductivity and Seebeck coefficients of the printed $SnSe_2$ samples were also measured in the temperature range of 300–550 K (Fig. 4d, f). The electrical conductivity of all samples increased with increasing temperature. Among the samples investigated, that annealed for 15 min showed the highest electrical conductivity of 279 S $m^{-1}$ at room temperature (Fig. 4d). This value is ~30% of the electrical conductivity reported for undoped high-quality $SnSe_2$ single-crystal bulk samples grown by the temperature gradient method, which indicates the high quality of our samples fabricated by the DLP method[63]. The Seebeck coefficients of the samples ranged from −290 to −194 μV $K^{-1}$ (Fig. 4f), which are slightly lower than those observed in $SnSe_2$ single crystals; such values may be attributed to the higher carrier concentration of the samples than $2.26 \times 10^{-18}$ $cm^{-3}$ of $SnSe_2$ single crystal since the Seebeck coefficient is inversely proportional to the carrier concentration[63,64]. The maximum power factor of the samples at 550 K was ~0.37 μW $cm^{-1}$ $K^{-2}$ (Fig. 4h). These results demonstrate the feasibility of our method for patterning highly functional crystalline metal chalcogenides without substantial losses in their intrinsic properties.

### Fabrication and evaluation of micro thermoelectric generator

We fabricated a micro-scale thermoelectric generator via the DLP printing of multiple p-type $Cu_2S$ and n-type $SnSe_2$ legs. DLP-based printing enables the simultaneous, rapid, and straightforward fabrication of several tens of thermoelectric legs of identical printing quality. We designed the shapes of the p-type and n-type legs to enable

their direct connection without the need for metal electrodes, which is beneficial for simplifying the manufacture of the devices. Using a digital mask composed of ten pairs of 300 μm-wide p-type and n-type legs (Supplementary Fig. 6g, h), we fabricated the desired patterns via a sequential patterning method. Here, the p-type $Cu_2S$ legs were patterned and heat-treated at 723 K, and the n-type $SnSe_2$ legs were patterned and annealed at 573 K (Fig. 5a). The power performance of the printed micro-scale thermoelectric generator was measured as each side was heated and cooled using a ceramic heater and Peltier cooler, respectively (Supplementary Fig. 18). During heating, the hot-side temperature gradually increased to 370 K, while the cold-side temperature remained at ~304 K (Supplementary Fig. 19). As the measured temperatures increased, the output voltage and power increased linearly and quadratically, respectively, demonstrating reliable power generation (Fig. 5b). At a maximum temperature difference of 65 K, the maximum output voltage and power density of the micro-scale thermoelectric generator reached 223.5 mV and 0.564 mW $cm^{-2}$, respectively (Fig. 5c).

We further fabricate the cross-plane thermoelectric device by the DLP printing process. The schematic illustration (Supplementary Fig. 20a) shows the entire fabrication process, in which we used one pair of the DLP-printed $Cu_2S$ and $SnSe_2$ films as p-type and n-type thermoelectric semiconductors. Since the current printing process is not applicable to electrode materials, we used the deposited Au and Ag paste layers as the bottom and top electrodes, respectively. The device resistance of ~150 Ω was observed at room temperature. We measured the power-generating performance of the fabricated generator by heating the bottom with a hot plate and air-cooling the top of the device. Because the $Cu_2S$ and $SnSe_2$ layers have an extremely thin thickness of ~500 nm, we couldn't obtain a temperature difference >1 K in the steady-state. However, the device showed an increase in the output voltages by two times and output powers by four times as the temperature difference increased from 0.2 K to 0.4 K, respectively, demonstrating the reliable power-generating performance (Supplementary Fig. 20b). Such power generation performance observed in the in-plane and cross-plane clearly demonstrates the potential of our DLP-based optical printing process for fabricating integratable micro-scale 2D or 3D devices with diverse heterostructured functional

materials. Indeed, our process may ultimately be able to facilitate the integration of electronic devices into other systems.

## Discussion

In summary, we established a generalised DLP-based optical printing technology to obtain diverse crystalline metal chalcogenides with 2D and 2.5D architectures by developing photocurable ChaM-based inorganic inks. The combination of mask-less DLP-based printing and our photocurable inorganic inks led to the facile fabrication of various inorganic materials with 2D and 2.5D architectures and excellent shape uniformity and fidelity. Our method features universal printability, including diverse functional metal chalcogenides and 2D layered transition-metal dichalcogenides. We demonstrated the feasibility of our approach by fabricating and evaluating a micro-scale thermoelectric device.

Unlike previous optical printing processes, our approach provides two-step patterning method that enables the direct use of inks, which could significantly reduce the manufacturing cost and time, and raises the possibility of building 3D architectures. The proposed process expands the availability of optically printable materials to inorganic semiconductors, which were previously limited to photocurable polymers and composites with inorganic fillers. We further summarised the recently reported printing methods for inorganic materials in terms of printable materials, printing speed, spatial resolution, processing steps, and the properties of materials (Supplementary Table 3). Our printing process are advantageous for the printing speed and simplicity in the printing process compared with other methods including inkjet printing, transfer printing, extrusion-based 3D printing, and other optical printing processes. We believe that our technology, in combination with high-resolution equipment, such as two-photon lithography, can realise the patterning of inorganic materials with excellent quality and resolution. Our DLP-based optical printing technology presents an alternative approach to photolithography and can serve as a platform technology for the cost-effective, simple, and high-resolution direct architecturing of inorganic materials.

## Methods

### Materials
Antimony(III) sulfide (99.995%. Sigma Aldrich), antimony(III) selenide (99.99%, Sigma Aldrich), tin powder (99.8%, 325 mesh, Acros Organics), selenium powder (99.999%, 200 mesh, Alfa Aesar), sulfur (99.998%, Sigma Aldrich), copper powder (99%, Alfa Aesar), ammonium sulfide solution (40–48 wt% in $H_2O$, Sigma Aldrich), chloroplatinic acid solution (8 wt% in $H_2O$, Sigma Aldrich), ammonium tetrathiomolybdate (99.97%, Sigma Aldrich), 2-[2-(5-methylfuran-2-yl) vinyl]-4,6-bis(trichloromethyl)-1,3,5-triazine (MFVT, TCI), *N*-(trifluoromethylsulfonyloxy)-1,8-naphthalimide (IM-NIT, WIMAS Corp.), ethylenediamine (≥99.5%, Sigma Aldrich), 1,2-ethanedithiol (≥98.0%, Sigma Aldrich), anhydrous acetonitrile (99.8%, Sigma Aldrich), anhydrous toluene (99.8%, Sigma Aldrich), *N*-methyl formamide (99%, Sigma Aldrich), anhydrous dimethyl sulfoxide (≥99.9%, Sigma Aldrich), ethyl alcohol (99.5%, Samchun), and methyl alcohol (99.5%, Samchun) were used in the experiments. All chemicals and reagents were used as received without further purification.

### Material Characterisation
**Scanning electron microscopy (SEM).** The shapes and microstructures of the various printed metal chalcogenides were obtained using a field-emission scanning electron microscope (FE-SEM Nano230) operated at 10 kV. Elemental analysis of the printed metal chalcogenides was performed by energy-dispersive X-ray spectroscopy (EDS) using the same SEM instrument.

**High-power X-ray diffractometry (XRD).** The XRD patterns of the printed metal chalcogenides were acquired using a D/MAX2500V/PC instrument (Rigaku) with a Cu-rotating anode X-ray source operated at 40 kV and 200 mA.

**UV-vis absorption spectroscopy.** The UV-absorption spectra of the photocurable ChaM-based inks and PAG solution were measured using a UV-vis spectrophotometer (UV-2600, Shimadzu) at room temperature.

**Confocal Raman spectroscopy.** The Raman spectra of the printed $MoS_2$ and $PtS_2$ layers were obtained using a confocal Raman microscope (Alpha300R, WITec) at room temperature with an excitation wavelength of 532 nm.

**Interferometric scattering-based 3D measurements.** The 3D scanned images, height profile, and roughness of the printed structures were obtained by an NV-3000 instrument (NanoSystem) using interferometric scattering.

**ζ-potential and dynamic light scattering (DLS).** The ζ-potential and DLS sizes of the photocurable ChaM-based inks were measured before and after UV irradiation using a Zetasizer Nano-ZS (Malvern).

**Optical microscopy (OM).** OM images of the printed layers were obtained using a semi-motorised fluorescence microscope BX53M (Olympus).

### Synthesis of the chalcogenidometallate solutions
**Sb-based chalcogenidometallates.** Sb-based ChaM solutions were synthesised using an amine/thiol alkahest solvent with additional purification. The overall synthetic process was performed in a $N_2$-filled glove box. In a typical experiment, 33.97 mg of $Sb_2S_3$ powder was added to a mixture of 1 ml of ethylenediamine and 0.1 ml of ethanedithiol (10:1, v/v) in a 10 ml vial with a stirring bar. The $Sb_2S_3$ powders were dissolved on a 50 °C hot plate in a glove box for 2.5 h with stirring until the powders were completely dissolved. Thereafter, the antisolvent acetonitrile (22 ml) was added to the $Sb_2S_3$ solution for purification (20:1, v/v). This mixture was centrifuged at $7142 \times g$ for 5 min to precipitate the purified ChaM precursors. The precipitate was redispersed in 1 ml of DMSO, added with 5 ml of acetonitrile (5:1, v/v), and centrifuged at $11,515 \times g$ for 5 min. Finally, the precipitate was dispersed in 0.5 ml of DMSO. Exactly 56.15 mg of $Sb_2Se_3$ powder was added to a mixture of ethylenediamine and ethanedithiol as described above. The $Sb_2Se_3$ powder was then dissolved for 1 h. Purification via the same method described above followed. Finally, the precipitate was redispersed in 1 ml of DMSO.

**Sn-based chalcogenidometallates.** To obtain SnS, we added Sn (110 mg) and S (30 mg) powders to a mixture of 1.6 ml of ethylenediamine and 0.16 ml of ethanedithiol (10:1, v/v) in a 10 ml vial with a stirring bar. The Sn and S powders were stirred for 12 h until the powders were completely dissolved and the solution became colourless. The anti-solvent acetonitrile was subsequently added to the SnS solution for purification (20:1, v/v), and the mixture was centrifuged at $7142 \times g$ for 5 min. The precipitate was redispersed in NMF, added with anhydrous toluene (3:1, v/v), and centrifuged at $11,515 \times g$ for 5 min for purification. Finally, the precipitate was dispersed in 1 ml of NMF. To obtain SnSe, we added 90 mg of Sn powder and 60 mg of Se powder to a mixture of 1.6 ml of ethylenediamine and 0.16 ml of ethanedithiol (10:1, v/v) in a 10 ml vial. The mixture was then dissolved for 1 h and 40 min. Purification via the same method employed for SnS followed. Finally, the precipitate was redispersed in 2 ml of NMF.

**Cu-based chalcogenidometallates.** $Cu_2Se$ powder was synthesised by the high-energy ball-milling of Cu and Se powders (2:1, atomic ratio) for 200 min as described in a previous report[25]. Ball-milled $Cu_2Se$

powder (103 mg) was added to a mixture of 1 ml of ethylenediamine and 0.1 ml of ethanedithiol (10:1, v/v) in a 10 ml vial with a stirring bar. The $Cu_2Se$ powders were mixed for 45 min until they were completely dissolved. The anti-solvent acetonitrile (5.5 ml) was added to the $Cu_2Se$ solutions (5:1, v/v) to purify the synthesised ChaM. This mixture was centrifuged at $7142 \times g$ for 5 min to precipitate the purified ChaM precursors, which were then redispersed in 1 ml of DMSO. This purification step was repeated twice.

**Ammonium thioplatinate.** Ammonium thioplatinate was synthesised using a modified method based on a previous report[39]. Exactly 3 g of S powder was added to 10 ml of an ammonium sulfide/distilled water solution (5 ml each). The mixture was stirred at room temperature until its colour turned red, signifying the formation of ammonium polysulfides. Then, 1.25 ml of chloroplatinic acid solution was slowly dropped into this red solution. The solution was stirred overnight and changed in colour from black-red to orange. The precipitate was then centrifuged at $5,725 \times g$ for 3 min. The precipitates were purified thrice with methanol and toluene (1:14, v/v) by centrifugation. Ammonium thioplatinate was redispersed in NMF at a concentration of 10 mg/ml.

**Ammonium tetrathiomolybdate.** Ammonium tetrathiomolybdate $((NH_4)_2MoS_4)$ (90 mg) was dissolved in 2 ml of NMF (2 ml; 45 mg/ml).

The completely dissolved inorganic solutions maintained their solubility for several months without any precipitation.

## DLP-based optical printing process

**Photocurable inorganic inks.** 2-[2-(5-Methylfuran-2-yl)vinyl]-4,6-bis(trichloromethyl)-1,3,5-triazine (MFVT) and N-(trifluoromethyl-sulfonyloxy)-1,8-naphthalimide (IM-NIT) were dissolved in several organic solvents, including ethanol, acetonitrile, and DMSO. The solubility limits of PAG differed for each solvent. In our experiments, (we used 0.02 M PAG-acetonitrile and 0.02 M) PAG-DMSO. Photocurable inorganic inks of the metal chalcogenides were prepared using a combination of ChaM-based inks and PAG solutions. The most important aspect of the DLP-based printing process is that the solubility of the ChaM-based inks–PAG mixture must be well maintained without any precipitation prior to UV exposure. Therefore, optimisation of the ChaM-based ink/PAG ratio is a key element for DLP-based printing. After a careful analysis of the detailed reactions of each photocurable inorganic ink, $Sb_2S_3$ (DMSO)/PAG-MFVT (0.02 M, DMSO) (1:3 v/v), $Sb_2Se_3$ (DMSO)/PAG-IM-NIT (0.02 M, DMSO) (1:1 v/v), SnS (NMF)/PAG-IM-NIT (0.02 M, ACN) (1:1.5 v/v), SnSe (NMF)/PAG-IM-NIT (0.02 M, DMSO) (1:1.25 v/v), $Cu_2S$ (DMSO)/PAG-MFVT (0.02 M, DMSO) (1:3 v/v), $MoS_2$ (NMF)/PAG-MFVT (0.02 M, ACN) (1:2.5 v/v), $PtS_2$ (NMF)/PAG-MFVT (0.02 M, ACN) (1:1.25 v/v) were finally prepared. These conditions can be changed according to various ink parameters, such as the ChaM solution concentration, PAG concentration, type, and solvent.

**DLP-based optical printing.** Two DLP systems were used for DLP-based optical printing. The first system uses a commercial DLP 3D printer (Nobel Superfine, XYZ Printing), which can provide 405 nm UV LED illumination over an area of 64 mm × 120 mm with a minimum X-Y resolution of 50 μm. This printer could produce sliced stl image files using XYZware Novel software. Because this commercial DLP 3D printing system is installed in a UV-protected $N_2$-filled glovebox, air-sensitive photocurable inorganic inks can be printed on this system. The second system uses hand-crafted DLP equipment (Luxbeam Rapid System, Visiotech). This system can illuminate a maximum area of 5 mm × 3 mm with 365 nm UV light. The maximum power and minimum resolution were 4 W and ~10 μm, respectively. The automation programme for optical printing uses g-code-based Visual Studio with sliced 3D images. The 3D architectures were obtained via a Z-stage (Linax, Lxc), which is capable of controlled 100 nm-scale

movement, and the light exposure was adjusted by a shutter (Uniblitz, VED24).

## Heat treatment of the printed metal chalcogenides

Antimony sulfide ($Sb_2S_3$) was annealed at 573 K for 1 min in ambient sulfur. Antimony selenide ($Sb_2Se_3$) was obtained by heat treatment at 573 K for 1 min. Tin sulfide (SnS) was recrystallised at 623 K for 1 min. Tin diselenide ($SnSe_2$) was annealed at 573 K for 10 min, and tin selenide (SnSe) was annealed at 723 K for 3 min. Copper sulfide ($Cu_2S$) was obtained by heat treatment at 723 K for 10 min. The heat treatment of the ChaM-based printed layers was conducted in a $N_2$-filled glovebox. Molybdenum disulfide ($MoS_2$) was obtained via a two-step heat treatment. After 5 min of annealing at 573 K in ambient sulfur, a stable $MoS_3$ phase was formed. Additional annealing at 723 K in 15% $H_2$ was subsequently conducted in a tube furnace for 10 min. Platinum disulfide ($PtS_2$) was subjected to a two-step heat treatment. The pre-heating treatment was conducted at 873 K for 1 h in a tube furnace under a $N_2$ atmosphere. The printed layers were annealed at 773 K for 30 min in ambient sulfur in a $N_2$-filled glove box.

## Thermoelectric properties of the printed layers

The Hall effect was measured at room temperature using a Hall measurement system (HMS-5000, ECOPIA). The charge carrier mobilities of the printed $Cu_2S$ and $SnSe_2$ were calculated using the obtained Hall coefficients and electrical conductivities measured by the Van der Pauw method. The temperature-dependent electrical conductivity and Seebeck coefficients of the printed $Cu_2S$ and $SnSe_2$ were measured using thermal analysis equipment (SBA 458 Nemesis, Netzsch) under inert conditions in the temperature range of 300–600 K. The typical size of a test specimen was 1.2 cm × 1.2 cm.

## Fabrication and evaluation of in-plane micro-scale thermoelectric generator

Approximately 600 μL of photocurable $Cu_2S$-based ink was poured on a glass substrate in a ink bath to print p-type thermoelectric legs. A digital mask with 300 μm-scale thermoelectric legs was exposed to UV light from the DLP 3D printer. The printed p-type $Cu_2S$ legs were rinsed with the same polar solvent and annealed at 723 K for 15 min on a hot plate in a $N_2$-filled glovebox. Approximately 600 μL of photocurable SnSe-based ink was poured onto a p-type leg-patterned glass substrate to obtain n-type thermoelectric legs. The same procedures were performed using a modified n-type thermoelectric leg digital mask. The printed n-type $SnSe_2$ legs were rinsed and annealed at 573 K for 15 min. Ten pairs of p- and n-type thermoelectric legs were securely connected. To measure the power generation of the fabricated micro-scale thermoelectric generator, we connected two K-type thermocouples to a Keithley 2000 multimeter and the edges of the thermoelectric legs on the hot (i.e. on the ceramic heater) and cold (i.e. on the Peltier cooler) sides of the generator. The cold-side temperature was maintained at ~304 K, whereas the hot-side temperature was gradually increased to 370 K. The micro thermoelectric generator was connected to a Keithley 2400 instrument, and the output voltages were measured for five temperature-difference cases with an applied sweep current. The output power $P$ was calculated using the formula $P = V \times I$, where $V$ is the voltage and $I$ is the current.

## Fabrication and evaluation of cross-plane thermoelectric generator

The patterned bottom electrodes (Cr 5 nm/ Au 80 nm) were thermally deposited on a 1.5 cm × 1.5 cm glass substrate. Prepatterned substrate was oppositely put in the photocurable $Cu_2S$-based inks, and UV light was exposed from the DLP 3D printer. The printed p-type $Cu_2S$ leg was rinsed with the same polar solvent and annealed at 723 K for 15 min on a hot plate in a $N_2$-filled glovebox. The same procedures were performed using the photocurable SnSe-based inks. The printed n-type $SnSe_2$ leg

was rinsed and annealed at 573 K for 15 min. An insulating layer (poly-imide tape) was introduced between the p-type and n-type legs. The top electrode (Ag paste) was deposited on the insulating layer, then one pair of p-type and n-type thermoelectric legs were electrically connected. The power-generating performance of the fabricated generator was measured by heating the bottom with a hot plate and air-cooling the top of the device. Temperature difference was measured with two K-type thermocouples to a Keithley 2000 multimeter, and output voltage was measured by a Keithley 2400 multimeter with an applied sweep current. The output power $P$ was calculated using the formula $P = V \times I$, where $V$ is the voltage, and $I$ is the current.

## Data availability
The data that support the findings of this study are available within the article and its Supplementary Information files. The source data underlying the figures of the Main Text are provided within the "Source Data" file. All raw data generated during the current study are available from the corresponding authors upon request. Source data are provided with this paper.

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

## Acknowledgements

This work was supported by the National Research Foundation of Korea (NRF) grant funded by the Korea government (the Ministry of Science and ICT) (NRF-2018M3A7B8060697, NRF-2022R1A2C3009129, NRF-2020M3D1A1110502, and NRF-2022R1A2C2008120), and the UNIST Research Fund (1.220024.01).

## Author contributions

S.B., H.W.B., and S.J. contributed equally to this work. S.B., H.W.B., S.J., J.L., and J.S.S. designed the experiments, analysed the data, and wrote the paper. S.B., H.W.B., D.H.G., W.C., Y.E.P and S.C. carried out the synthesis and basic characterization of materials. S.B., H.W.B. and S.J. performed the printing of metal chalcogenides. S.H.H. performed the characterisation of thermoelectric properties. S.B., J.Y., and M.K.C. fabricated and evaluated the thermoelectric devices. All authors discussed the results and commented on the manuscript.

## Competing interests

The authors declare no competing interests.
