## [Peer Review File · Nature Communications]

Generalised optical printing of photocurable metal chalcogenidesReviewer #1 (Remarks to the Author):

The authors introduce a technique for optical 3D printing of metal chalcogenide-based on a photoacid generator. The advantage of this technology compared with other optical patterning techniques is that it does not involve the use of a photo-sensitive polymeric matrix to achieve crosslinking during printing. This prevents the presence of organic residues that degrade the performance of the final material. This is a general method of interest for a broad range of fields related to microfabrication and electrical engineering. The work supports the claims with an adequate amount of results meeting the expected high standard of this journal in terms of presentation and explanation. However, I propose some modifications which, in my opinion, will improve the clarity and completeness of the report:

Line #26: Although the cross-section of the printed shapes can be altered throughout the thickness direction (Fig. 2f), the authors should be more conservative when claiming 3D printing. I think the process is potentially compatible with 3D printing but the thickness demonstrated corresponds still to the thin-film domain (0.5 μm). Also, the TEG demo is in-plane. Therefore, no real 3D architecture has been shown. The claim of 3D printing should be replaced with "compatibility with 3D printing" or "potential for 3D printing", etc.

Line #43: Since the authors have similar papers based on DIW, it would be relevant to introduce here a brief comparison between these two 3D printing methods.

Lines #54-56: Can you provide numbers for the efficiency? what about porosity?

Line #86: Why these materials? Why not Sb_2Te_3

Line #121: what alcoholic solvent is this one?

Line #173: see the comment from line #26.

Line #195: What is the thickness of the samples for XRD?

Line #196: The patterns are quite noisy and it is difficult to compare with the pdf card. I see clear texturing for 3c but not really for 3d, or at least not much compared to other panels such as 3f. Can the authors clarify more this point by improving the quality of the patterns or adding extra labels? A comparison in the SI between the expected peaks for textured vs non-textured materials and some details about the crystalline structure would be highly appreciated for clarity.

Line #197: Those peaks are labeled only for Fig. 3c but not for 3d.

Line #214: Do you mean respectively (meaning Cu_2s was annealed at 732 K and SnSe_2 at 573K)?

Line #216: Can you add evidence of this sentence?: "The annealing temperatures were selected after considering the thermal stability of the materials to conserve their stoichiometric composition 216 and microstructural integrity".

Line #222: Could the authors clarify how the evaporation of S leads to fewer Cu vacancies?

Line #226: Reducing pinholes and cracks is no evidence of reducing Cu vacancy point defects. They belong to different size scales. When talking about Cu vacancies, I can imagine atomic scale, whereas when referring to cracks and pinholes, we are talking about hundreds of nanometers or a few microns scale.

Line #253-254: This is a bold statement as fluctuations appear also in the SnSe_2

material and are likely due to typical experimental errors when measuring conductivity and Seebeck. The authors should remove this claim or include further studies such as DSC measurements to support it.

Line #260-261: can the author clarify why materials with high carrier concentration show an increase of S with temp? A reference would be useful too.

Line #274: Can the authors elaborate on this statement. According to the Pisarenko relation, S should go up for lower n for highly doped semiconductors, which are usually employed for thermoelectrics (DOI: 10.1039/C1EE02612G). Is this what the authors mean? In the reference provided, the opposite is claimed because the authors of the referred paper compared an intrinsic semiconductor with a doped one.

Line #298: 3D devices are only potentially possible, not demonstrated.

Line #311: I suggest changing for 3-steps (or 2-steps if sintering is considered as a post-processing step) as exposure, and removal of green material are needed.

Line #315: a "to" is needed between "limited" and "photocurable".

Figure 2: Some of these panels do not provide any extra relevant info (a,b, and c show similar results, d and e also). Therefore they are redundant and they could be removed or transferred to the SI to make the main text more focused.

Figure 4: Can the authors provide details on how they fit the experimental points to the dashed curves? Did they use a mathematical function? if so, which one? Or did they use any software option?

Reviewer #2 (Remarks to the Author):

The manuscript reported a very interesting concept of optical printing of photocurable metal chalcogenides. Optical printing has been mostly used for printing polymers or their composites. The concept reported in this work opens new opportunities of using optical printing to print inorganic semiconductors. I would like to recommend the publication of this article after the authors address the following comments:

1. The reported charge carrier mobility and conductivity seems still considerably lower than the best reported values of the same materials made by other techniques. The authors should provide some explanations why the property is still relatively inferior using the optical printing method.

2. There are many printing methods available to print the inorganic inks. I suggest the authors provide a performance comparison of the optical printing vs. other printing methods in terms of printing speed, spatial resolution, properties of printed materials, etc.

3. The authors demonstrated an in plane TE device. The optical printing method seems a very good method to fabricate cross-plane TE devices. The cross-plane TE devices are much more applicable for energy harvesting devices than in plane TE devices. If possible, the authors may attempt to demonstrate some cross-plane TE devices, which can make this work even more appealing.

Response to the reviewers' comments

The followings are the responses to the reviewers' comments for the manuscript "Generalised optical printing of photocurable metal chalcogenides."

▪ **Reviewer #1**

General comment: The authors introduce a technique for optical 3D printing of metal chalcogenide-based on a photoacid generator. The advantage of this technology compared with other optical patterning techniques is that it does not involve the use of a photo-sensitive polymeric matrix to achieve crosslinking during printing. This prevents the presence of organic residues that degrade the performance of the final material. This is a general method of interest for a broad range of fields related to microfabrication and electrical engineering.

The work supports the claims with an adequate amount of results meeting the expected high standard of this journal in terms of presentation and explanation. However, I propose some modifications which, in my opinion, will improve the clarity and completeness of the report.

Response: We appreciate the reviewer's valuable time and effort in evaluating our manuscript. We truly agree with the reviewer's comments about the need for the additional characterisation and clarifications of the process and materials. We sincerely addressed all the comments and believe that these revisions made a significant improvement in the quality of the manuscript. We thank the reviewer again for the positive comment.

Comment 1: Line #26: Although the cross-section of the printed shapes can be altered throughout the thickness direction (Fig. 2f), the authors should be more conservative when claiming 3D printing. I think the process is potentially compatible with 3D printing but the thickness demonstrated corresponds still to the thin-film domain (0.5 μm). Also, the TEG demo is in-plane. Therefore, no real 3D architecture has been shown. The claim of 3D printing should be replaced with "compatibility with 3D printing" or "potential for 3D printing", etc.

Response: We appreciate the reviewer's valuable comment. We agree with the reviewer's comment that the claim of "3D printing" somewhat exaggerated our printing process in this study. To overcome

this issue, we have tried the optical “3D printing” of metal chalcogenides with a higher aspect ratio by our process, where the micrometre-scale architectures were fabricated by the layer-by-layer DLP printing. As shown in the SEM images and the height profile (Fig. A1), we printed the 1.5 μm -thick pyramidal 3D architecture constructed by thirty layers of MoS_2 -based ChaMs. These pyramids consist of three squares with different lengths of 500, 400, and 300 μm . We believe that the realisation of the printing of micrometer-thick architectures clearly demonstrates the compatibility of our approach to 3D printing, even though this demonstration does not sufficiently validate the concept of 3D printing in the current status.

Fig. A1 (a) SEM image and (b) height profile of the DLP-printed pyramidal architectures (Scale bar: 500 μm).

Accordingly, we replaced the claim of “3D printing” were replaced with “2.5D or compatibility with 3D printing”. Also, we included the data of the pyramid architectures (Fig. A1) and the related discussion in the revised manuscript and Supplementary Information (page 9, Supplementary Fig. 10).

“In addition, the micrometre-scale architectures were fabricated by the layer-by-layer DLP printing. As shown in the SEM images and the height profile (Supplementary Fig. 10), we printed the 1.5 μm -thick pyramidal 3D architecture constructed by thirty layers of MoS_2 -based ChaMs. These pyramids consist of three squares with different lengths of 500, 400, and 300 μm . These results demonstrate the compatibility of our approach to 3D printing of inorganic metal chalcogenides.”

Comment 2: Line #43: Since the authors have similar papers based on DIW, it would be relevant to introduce here a brief comparison between these two 3D printing methods.

Response: We appreciate the reviewer's fruitful comment. As the reviewer commented, our group have studied the DIW 3D printing process for fabricating various types of inorganic electronic materials. The DIW is an extrusion-based 3D printing method to create meso- and micro-scales architectures. In the DIW, the liquid-phase "ink" is dispensed out of small nozzles under controlled flow rates and deposited along digitally defined paths to fabricate 3D structures layer-by-layer^{A1}. Recently, **by developing the viscoelastic inks containing functional inorganics, this method has been extensively utilized for fabricating various types of functional inorganic materials. This is the stark contrast to the optical 3D printing techniques such as digital light processing (DLP) and stereolithography, which suffer from the critical issue of limited printable materials,** thus their applications in the electronic and energy fields are restricted.

Accordingly, we included the following sentences on the revised manuscript (page 3).

"This is the stark contrast to the direct ink writing (DIW) techniques, which is an extrusion-based 3D printing method to create meso- and micro-scales architectures. By developing the viscoelastic inks containing inorganic particles, the DIW has been extensively utilized for fabricating various types of functional inorganic materials¹⁸⁻²⁰. For example, the DLP-printed thermoelectric BiSbTe materials was reported to exhibit the ZT value of 0.12, which is one order of magnitude lower than that of inorganic bulk reference materials or that of even DIW-printed BiSbTe because the printed objects contain organic residues²¹."

Comment 3: Lines #54-56: Can you provide numbers for the efficiency? what about porosity?

Response: We appreciate the reviewer's careful comment. In the Ref. 18 in the revised manuscript, **the maximum ZT value of the printed composite was 0.12 at 43 °C,** which was obtained in the BiSbTe-based composites with **a porosity of 62%.**

Accordingly, we replaced the previous sentence with the following one in the revised manuscript (page 3,4).

"For example, the DLP-printed thermoelectric BiSbTe materials was reported to exhibit the ZT value of 0.12, which is one order of magnitude lower than that of inorganic bulk reference materials or that of even DIW-printed BiSbTe because of the printed objects contain organic residues²¹."

Comment 4: Line #86: Why these materials? Why not Sb₂Te₃.

Response: We appreciate the reviewer's thoughtful comment. **Our group reported soluble telluride-based molecular Sb₂Te₃ precursors, which can be used as the inks for the current DLP printing**^{A2}. The Sb₂Te₃ precursor was synthesized by the multi-step processes of the synthesis of polymeric Sb₂Te₃ precursor and the subsequent superhydride and tri-n-octylphosphine (TOP) treatments (Fig. A2). Although this final precursor could generate the high-quality thin films by the spin coating process, **the complicated synthetic route for the Sb₂Te₃ precursor requires tight engineering in every step for ensuring the composition and purity of precursors. We have tried to use this precursor for the optical printing process but we concluded that this precursor was not suitable for the current optical printing process because the mixing step with the photoacid generator (PAG) to provide the photoreactivity was found to easily degrade the composition of the Sb₂Te₃ precursor. Moreover, the limited solubility of the Sb₂Te₃ precursor in the DMSO and acetonitrile made it difficult to use this precursor in the current optical printing process that requires the relatively high concentration of precursor.** Accordingly, we chose the Cu₂S and SnSe₂ as thermoelectric materials for fabricating power generating devices.

Fig. A2 Scheme of the procedures for the molecular Sb₂Te₃ precursors and thin film fabrication (adapted from Ref. A2).

Comment 5: Line #121: what alcoholic solvent is this one?

Response: We apologize for the confusing information in the previous manuscript. Photoacid generator (PAG) is a compound that formed the photoproducts included acidic species through reaction or dissociation upon illumination. They can be dissociated via radical pathways or cleavage of C-O, S-O and N-O bonds, requiring hydrogen extractions from surrounding compounds or solvents like alcohols. In addition to alcohols, diverse ranges of solvents can be used as the proton source for activating the PAG. **In this study, we have tested diverse ranges of solvents such as alcoholic solvent (ethanol,**

methanol, isopropanol, etc.) and non-alcoholic solvent (DMSO, acetonitrile, etc.).

Accordingly, we included the following phrase on the revised manuscript (page 6).

“alcoholic solvent (ethanol, methanol, isopropanol, etc.) and non-alcoholic solvent (DMSO, acetonitrile, etc.)”

Comment 6: Line #173: see the comment from line #26.

Response: Please refer to the Response for the Comment 1.

Comment 7: Line #195: What is the thickness of the samples for XRD?

Response: We measured the thickness of the samples for the XRD analysis. As shown in Fig. A3, **the thickness of the sample is about 250 nm.**

Fig. A3 3D scan height profile of the printed samples for XRD analysis.

We included the thickness profile of printed samples for XRD (Fig. A3) on the revised Supplementary Information and the following phrase in the revised manuscript (page 10, Supplementary Fig. 12).

“with the thickness of 250 nm (Supplementary Fig. 12)”

Comment 8: Line #196: The patterns are quite noisy and it is difficult to compare with the pdf card. I see clear texturing for 3c but not really for 3d, or at least not much compared to other panels such as 3f. Can the authors clarify more this point by improving the quality of the patterns or adding extra labels? A comparison in the SI between the expected peaks for textured vs non-textured materials and some details about the crystalline structure would be highly appreciated for clarity.

Response: We fully agree with the reviewer’s comment that the XRD pattern of SnSe₂ (Fig. 3d) is quite noisy and difficult to compare with the pdf card. As the reviewer requested, **we added the extra labels in the XRD pattern of printed SnSe₂** (Fig. A4a). In addition, **we newly obtained the XRD pattern of non-textured SnSe₂**. This sample was the photocured SnSe₂ powders which was heated under the identical heat treatment condition for the textured thin film. **The XRD pattern of the photocured SnSe₂ powders (Fig. A4b) corresponds to the hexagonal SnSe₂ phase reference and shows the randomly oriented pattern. This is in contrast to that of the patterned SnSe₂, in which the *a*-axis peaks of the (001), (003), and (004) are significantly pronounced compared with other peaks.** These results demonstrate the crystallographic textures in the printed SnSe₂ layer.

Fig. A4 XRD pattern of (a) the printed SnSe₂ and (b) photocured SnSe₂ powder.

Accordingly, we replaced the previous Fig. 3d with the newly obtained XRD data (Fig. A4a) and included the XRD pattern of non-textured SnSe₂ (Fig. A4b) in the revised Supplementary Information (Supplementary Fig. 13) and the following sentence in the revised manuscript (page 10).

“Also, the SnSe₂ exhibited the a-axis peaks of the (001), (003), and (004) planes in the XRD pattern, showing a strong orientation when it compared with that of the non-textured SnSe₂ phase (Supplementary Fig. 13).”

Comment 9: Line #197: Those peaks are labeled only for Fig. 3c but not for 3d.

Response: Please refer to the Response for the Comment 8.

Comment 10: Line #214: Do you mean respectively (meaning Cu₂S was annealed at 732 K and SnSe₂ at 573K)?

Response: We apologize for the confusing sentence in the previous manuscript. **The printed 2D Cu₂S samples was annealed at 723 K and 2D SnSe₂ samples was annealed at 573 K.** To make it clear, we corrected the previous sentence on the revised manuscript, as follows (page 11).

“Here, we fabricated 2D Cu₂S and SnSe₂ samples using the DLP method and then annealed Cu₂S at 723 for 5, 10, and 15 min, and SnSe₂ at 573 K for 7, 10, and 15 min, respectively.”

Comment 11: Line #216: Can you add evidence of this sentence?: "The annealing temperatures were selected after considering the thermal stability of the materials to conserve their stoichiometric composition and microstructural integrity".

Response: We appreciate the reviewer’s thoughtful comment. To address this issue, we systematically annealed the SnSe₂ and Cu₂S samples at different temperatures and characterised them by the XRD and SEM analysis. The SEM images of SnSe₂ samples (Fig. A5) shows that the microstructures were not changed significantly. However, **the SnSe phase (p-type) was started to detected in the XRD patterns at 623 K and the SnSe₂ phase was fully transformed to the SnSe phase at 673 K.** This composition transition was well-known to be attributable to the evaporation of Se^{A3}. **Since SnSe crystal**

generally exhibit the p-type properties, we chose the annealing temperature of 573 K to conserve the n-type character of the SnSe₂ sample. On the other hand, regardless of the annealing temperatures, all Cu₂S samples shows the smooth film with good coverage, as shown in the SEM images (Fig. A6). However, the XRD patterns of the samples annealed at 623 K and 673 K corresponded to the Cu_{1.8}S bulk reference, while the samples annealed at higher temperatures showed the Cu_{1.96}S phase (Fig. A6b). In general, Cu_{1.96}S crystal is known to exhibit significantly higher thermoelectric properties than those of Cu_{1.8}S (Fig. A7)^{A4}. Accordingly, we chose the annealing temperature of 723 K to obtain the Cu_{1.96}S phase.

Fig. A5 (a) SEM images and (b) XRD patterns of the printed SnSe₂ according to the heat treatment temperature. (Scale bar: 1 μ m)

Fig. A6 (a) SEM images and (b) XRD patterns of the printed Cu₂S according to the heat treatment temperature. (Scale bar: 1 μ m)

Fig. A7 ZT values of Cu_{2-x}S pellets ($\text{Cu}_{1.96}\text{S}$ and $\text{Cu}_{1.8}\text{S}$) annealed under different atmospheres Ar/H_2 and Ar , respectively (adapted from Ref. A4).

Accordingly, we included the SEM and XRD data and the related discussion in the revised Supplementary Information (Supplementary Discussion and Supplementary Fig. 14 and 15).

“To optimise the annealing conditions, we systematically annealed the SnSe_2 and Cu_2S samples at different temperatures and characterised them by the XRD and SEM analysis. The SEM images of SnSe_2 samples (Supplementary Fig. 14) shows that the microstructures were not changed significantly. However, the SnSe phase (p-type) was started to detected in the XRD patterns at 623 K and the SnSe_2 phase was fully transformed to the SnSe phase at 673 K. This composition transition was well-known to be attributable to the evaporation of Se^{20} . Since SnSe crystal generally exhibit the p-type properties, we chose the annealing temperature of 573 K to conserve the n-type character of the SnSe_2 sample. On the other hand, regardless of the annealing temperatures, all Cu_2S samples shows the smooth film with good coverage, as shown in the SEM images (Supplementary Fig. 15). However, the XRD patterns of the samples annealed at 623 K and 673 K corresponded to the $\text{Cu}_{1.8}\text{S}$ bulk reference, while the samples annealed at higher temperatures showed the $\text{Cu}_{1.96}\text{S}$ phase (Supplementary Fig. 15b). In general, $\text{Cu}_{1.96}\text{S}$ crystal is known to exhibit significantly higher thermoelectric properties than those of $\text{Cu}_{1.8}\text{S}^{21}$. Accordingly, we chose the annealing temperature of 723 K to obtain the $\text{Cu}_{1.96}\text{S}$ phase.”

Comment 12: Line #222: Could the authors clarify how the evaporation of S leads to fewer Cu vacancies?

Response: We appreciate the reviewer’s constructive comment on our manuscript. Copper chalcogenides Cu_{2-x}X ($\text{X} = \text{S}, \text{Se}$ or Te) have been studied as promising thermoelectric materials. The electrical and functional properties of copper chalcogenides depend on not only the crystal structures but stoichiometry of composition due to the carrier concentration. **Because the Cu vacancy in copper chalcogenides act as the hole donor, the hole carrier concentration is generally increased with**

increasing the copper deficiency from Cu_2X to Cu_{2-x}X . Lin et al. reported the decreased carrier concentrations of Cu_2Se thin films with increasing the heat treatment temperatures^{A5}. The author claimed that the higher annealing temperature leads to the lower Se contents, as a result, which reduces the self-doping effect of the copper deficiency and decrease the hole concentration (Fig. A8). Likewise, in our printed Cu_2S samples, the annealing time dependences of carrier concentrations could result from the leaving chalcogen (in our case, the S evaporation) that reduces the self-doping effect caused by the Cu deficiency and decreases carrier concentrations.

Fig. A8 The room temperature carrier concentration of Cu_2Se thin film annealed at different temperature (adapted from Ref. A5).

Accordingly, we included the following reference (Ref.A5) and phrase in the revised manuscript (Ref. 53, page 11).

53. Lin, Z. *et al.* A Solution Processable High-Performance Thermoelectric Copper Selenide Thin Film. *Adv. Mater.* **29**, 1606662 (2017).

“increasing the copper deficiency from Cu_2X to Cu_{2-x}X ⁵³.”

Comment 13: Line #226: Reducing pinholes and cracks is no evidence of reducing Cu vacancy point defects. They belong to different size scales. When talking about Cu vacancies, I can imagine atomic scale, whereas when referring to cracks and pinholes, we are talking about hundreds of nanometers or a few microns scale.

Response: We agree with the reviewer’s comment that the formation of pinholes and cracks were not related to the Cu vacancy point defect. **Accordingly, we removed the previous sentences and included the following sentences in the revised manuscript (page 11).**

“Moreover, the SEM images of the samples (Supplementary Fig. 16) showed fewer pinholes and cracks in the microstructures of samples heated for longer durations, which further contributed to the increase of the carrier mobility.”

Comment 14: Line #253-254: This is a bold statement as fluctuations appear also in the SnSe₂ material and are likely due to typical experimental errors when measuring conductivity and Seebeck. The authors should remove this claim or include further studies such as DSC measurements to support it.

Response: We fully agree with the reviewer’s comment that the temperature-dependent thermoelectric properties of the SnSe₂ thin films are somewhat confused at low temperatures. **To understand this behaviour, we newly characterised temperature-dependent thermoelectric properties of the newly fabricated SnSe₂ films (Fig. A9), providing the reliable data of thermoelectric properties.** In these data, **the fluctuation at the temperature from 350 K - 450 K were not observed in both electrical conductivities and Seebeck coefficients of all SnSe₂ films annealed for 7, 10, and 15 min.** Moreover, the newly obtained electrical conductivity and Seebeck coefficients of all samples show almost identical values to the previous data at room temperature within the equipment errors. The samples annealed for longer duration times shows the higher electrical conductivity and lower Seebeck coefficients, which are attributable to the electron concentration.

Fig. A9 Temperature dependences of (a) electrical conductivity, (b) Seebeck coefficient, and (c) power factor of the printed SnSe₂ as functions of the heat treatment time.

Accordingly, we replaced the previous thermoelectric properties with newly measured data (Fig.

A9) in the revised manuscript (Fig. 4).

Comment 15: Line #260-261: can the author clarify why materials with high carrier concentration show an increase of S with temp? A reference would be useful too.

Response: The temperature dependence of the Seebeck coefficient is known to strongly affected by the carrier concentrations due to the strength of bipolar effect. The bipolar effect on the Seebeck coefficient could be expressed as shown in equation of,

$$S = \frac{S_h \times \sigma_h - |S_e \times \sigma_e|}{\sigma_h + \sigma_e}$$

where S_h and S_e are the Seebeck coefficient of holes and electrons, and σ_h and σ_e are the electrical conductivities of holes and electrons, respectively^{A6}. In this equation, the Seebeck coefficient is determined by the electron and hole contributions to the electrical conductivity and Seebeck coefficient.

In the material with a low concentration of majority carrier, the minority carrier effect becomes stronger with higher temperatures by the thermal excitation, consequently, the negative temperature dependence can be observed at high temperatures. On the contrary, the material with a high concentration of the majority carrier, the minority carriers affect less the electrical transport. Accordingly, these materials usually show the positive temperature dependence in a wide temperature range. This behaviour is generally reflected in the temperature-dependent Seebeck coefficients of materials, which shows the shift of the peak temperatures with the increase of carrier concentrations of materials (Fig. A10)^{A7, A8}.

Fig. A10 (a) Room temperature carrier concentration and mobilities of the materials. (b) Temperature dependent Seebeck coefficients (adapted from Ref. A7) (c) Calculated Seebeck coefficient as a function of temperature for CaMg_2Bi_2 crystals with different carrier concentrations (n) (adapted from Ref. A8).

Likewise, in our study, the shift of the peak temperature of Seebeck coefficients can be

understood by the reduced bipolar effect being attributed to the higher carrier concentrations.

Accordingly, we included the new following references (A7 and A8) and the following sentences in the revised manuscript (Ref. 61 and 62, page 14).

61. Yang, S. E. *et al.* Composition-segmented BiSbTe thermoelectric generator fabricated by multimaterial 3D printing. *Nano Energy* **81**, 105638 (2021).

62. Gong, J. J. *et al.* Investigation of the bipolar effect in the thermoelectric material CaMg_2Bi_2 using a first-principles study. *Phys. Chem. Chem. Phys.* **18**, 16566-16574 (2016).

“Generally, the contribution of minority carriers to the Seebeck coefficient is negative and become more significant at higher temperatures by the thermal excitation, eventually causing the negative temperature dependence of the Seebeck coefficients. However, in the material with a high concentration of the majority carriers, the minority carriers affect less the electrical transport by suppressing the bipolar effect. Accordingly, these materials usually show the positive temperature dependences in a wide temperature range. This behaviour is generally reflected in the temperature-dependent Seebeck coefficients of materials, which shows the shift of the peak temperatures with the increase of carrier concentrations of materials^{61,62}.”

Comment 16: Line #274: Can the authors elaborate on this statement. According to the Pisarenko relation, S should go up for lower n for highly doped semiconductors, which are usually employed for thermoelectrics (DOI: 10.1039/C1EE02612G). Is this what the authors mean? In the reference provided, the opposite is claimed because the authors of the referred paper compared an intrinsic semiconductor with a doped one.

Response: We appreciate the reviewer’s valuable comment. As the reviewer commented, we intended to mention that the Seebeck coefficient rise up for lower carrier concentration for highly doped semiconductor since our samples have higher carrier concentration than that of the reported SnSe_2 single crystal. **As the reviewer requested, we replaced the previous sentences with the following sentences and cited the reference of A9 (DOI: 10.1039/C1EE02612G) that the reviewer suggested in the revised manuscript (page 14, Ref. 64).**

“The Seebeck coefficients of the samples ranged from -290 to $-194 \mu\text{V K}^{-1}$ (Fig. 4f), which are slightly lower than those observed in SnSe_2 single crystals; such values may be attributed to the higher carrier concentration of the samples than $2.26 \times 10^{-18} \text{ cm}^{-3}$ of SnSe_2 single crystal since the Seebeck coefficient is inversely proportional to the carrier concentration^{63,64}.”

Comment 17: Line #298: 3D devices are only potentially possible, not demonstrated.

Response: Please refer to the Response for the Comment 1.

Comment 18: Line #311: I suggest changing for 3-steps (or 2-steps if sintering is considered as a post-processing step) as exposure, and removal of green material are needed.

Response: As the reviewer suggested, we changed the expression of ‘single step’ with ‘two-step’ in the revised manuscript.

Comment 19: Line #315: a "to" is needed between "limited" and "photocurable".

Response: Thank you for the kind comment. We edited all grammatical errors and typos in the revised manuscript.

Comment 20: Figure 2: Some of these panels do not provide any extra relevant info (a,b, and c show similar results, d and e also). Therefore they are redundant and they could be removed or transferred to the SI to make the main text more focused.

Response: We generally agree with the reviewer’s comment that Fig. contains redundant panels. We moved the Fig. 2e to the Supplementary Information. However, Fig. 2a, b, and c provide the different information; e.g. Fig. 2a shows the spatial resolution of our DLP-based printing method, Fig. 2b shows the various shapes of printed patterns circles, squares, and triangles, which demonstrates the fidelity of the printed material at edges and corners. Moreover, Fig. 2c demonstrates high-throughput fabrication at once in a large area by our printing method. Thus, we would like to keep Fig. 2a-2c in the main Figure.

We replaced the previous Fig. 2 with the revised version (Fig. A11) in the revised manuscript.

Fig. A11 (a) SEM image of printed line patterns of PtS₂-based ChaM ink with widths ranging from 100 μm to 25 μm. (b) SEM image of the printed circles, squares, and triangles of PtS₂-based ChaM ink at different scales of tens to hundreds of micrometres. (c) SEM image of hundreds of printed square array patterns with a width of 100 μm at a scale of several millimetres. (d) SEM image of the printed geometric pattern of MoS₂-based ChaM ink with a linewidth of 50 μm created by the DLP method. (e) SEM and 3D scan analysis (inset) images of the printed 2.5D architectures composed of circle- and triangle-shaped layers. (f) Height profile of the printed 2.5D architecture. All scale bars in the panels a-e indicate 500 μm.

Comment 21: Figure 4: Can the authors provide details on how they fit the experimental points to the dashed curves? Did they use a mathematical function? if so, which one? Or did they use any software option?

Response: We plotted the dashed curves using the Origin software option, which calculates the polynomial fitting based on the experimental points. For example, the calculated polynomial curves could be expressed as shown in the equation of

$$y = Ax^2 + Bx + C$$

, where A and B are calculated constants based on the experimental points, and C is the intercept. However, we thought the polynomial fitting seems confusing in some data. Accordingly, we removed this fitting and added the simple B-spline lines on the data plots (Fig. A12) in the Origin software in the revised manuscript (Fig. 4).

Fig. A12 Charge carrier mobility and concentration of the printed (a) Cu₂S and (b) SnSe₂ as functions of the heat treatment time as detected by Hall effect measurements. Temperature dependences of (c, d) electrical conductivities, (e, f) Seebeck coefficients, and (g, h) power factors of the printed Cu₂S (c, e, g) and SnSe₂ (d, f, h) as functions of the heat treatment time.

References

- A1. Kim, F. *et al.* Direct ink writing of three-dimensional thermoelectric microarchitectures. *Nat. Electron.* **4**, 579-587 (2021).
- A2. Jo, S. *et al.* Soluble Telluride-Based Molecular Precursor for Solution-Processed High-Performance Thermoelectrics. *ACS Appl. Energy Mater.* **2**, 4582-4589 (2019).
- A3. Heo, S. H. *et al.* Composition change-driven texturing and doping in solution-processed SnSe thermoelectric thin films. *Nat. Commun.* **10**, 864 (2019).
- A4. Li, M. *et al.* Effect of the Annealing Atmosphere on Crystal Phase and Thermoelectric Properties of Copper Sulfide. *ACS Nano* **15**, 4967-4978 (2021).
- A5. Lin, Z. *et al.* A Solution Processable High-Performance Thermoelectric Copper Selenide Thin Film. *Adv. Mater.* **29**, 1606662 (2017).
- A6. Suh, D. *et al.* Enhanced thermoelectric performance of Bi_{0.5}Sb_{1.5}Te₃-expanded graphene composites by simultaneous modulation of electronic and thermal carrier transport. *Nano Energy* **13**, 67-76 (2015).
- A7. Yang, S. E. *et al.* Composition-segmented BiSbTe thermoelectric generator fabricated by multimaterial 3D printing. *Nano Energy* **81**, 105638 (2021).
- A8. Gong, J. J. *et al.* Investigation of the bipolar effect in the thermoelectric material CaMg₂Bi₂ using a first-principles study. *Phys. Chem. Chem. Phys.* **18**, 16566-16574 (2016).
- A9. Heremans, J. P., Wiendlocha, B. & Chamoire, A. M. Resonant levels in bulk thermoelectric semiconductors. *Energy Environ. Sci.* **5**, 5510-5530 (2012).

▪ Reviewer #2

General comment: The manuscript reported a very interesting concept of optical printing of photocurable metal chalcogenides. Optical printing has been mostly used for printing polymers or their composites. The concept reported in this work opens new opportunities of using optical printing to print inorganic semiconductors. I would like to recommend the publication of this article after the authors address the following comments:

Response: We appreciate the reviewer's valuable time and effort in evaluating our manuscript. We truly agree with the reviewer's comments about the need for the comparative discussion on the process and materials, and the realisation of the cross-plane thermoelectric device. We sincerely addressed all the comments and believe that these revisions made a significant improvement in the quality of the manuscript. We thank the reviewer again for the positive comment.

Comment 1: The reported charge carrier mobility and conductivity seems still considerably lower than the best reported values of the same materials made by other techniques. The authors should provide some explanations why the property is still relatively inferior using the optical printing method.

Response: We appreciate the reviewer's thoughtful comment. **We compared the carrier mobility and electrical conductivity of our printed Cu₂S and SnSe₂ samples with the reported properties of corresponding materials of bulk crystals, solution-processed films, and printed samples.** As shown in the **Table B1, we would like to emphasize that the properties of the reported solution-processes thin films or the 3D-printed samples are comparable to those of our samples,** even though the electrical conductivities and mobilities of our samples were roughly an order of magnitude lower than the bulk values.

Generally, the ink-processed materials such as thin or thick films fabricated by various coating or printing processes exhibit inferior properties to the bulk values. This can be understood with consideration that **these materials are usually more defective and less dense and include smaller grains compared with bulk materials synthesized by energy-intensive methods under harsh condition.** For example, **the grain sizes of our Cu₂S and SnSe₂ samples range in the several tens to several hundreds of nanometres, which are several orders of magnitudes lower than those observed in the reported bulks^{B2,B8,B9}.** These smaller grains should cause the higher density of

interfaces that can hinder the charge carrier transport. Moreover, **various types of defects can be formed in these materials since the inks are the mixtures of organic solvents, precursors, and sometimes some binders.** In our samples, the precursors are organometallic complexes of chalcogenidometallates with counter cation of ethylenediammonium, which are synthesized with ethylenediamine and ethanedithiol solvent mixtures. Thus, there can be many undesired impurities residues in the fabricated thin films, which can reduce the charge carrier mobility and electrical conductivity by the impurity scattering. Finally, **the ink-processed thin films generally have porous features in the microstructures;** e.g. the SEM images of our samples shows multiple pinholes in the nanometre scale. The formation of these pores is usually inevitable since they are created during the solvent drying or precursor decomposition that releases gases. All these reasons can attribute the reduced charge carrier mobility and resulting electrical conductivity.

Table B1. Comparison of the printed materials in this study with the corresponding reported materials fabricated by the different methods.

Ref	Materials	Properties (n : carrier concentration μ : carrier mobility σ : electrical conductivity)	Sample type and fabrication method
[B1]	Cu _{2-x} S	n : $7.27 \times 10^{20} \text{ cm}^{-3}$ at RT (Cu _{1.97} S) μ : $0.69 \text{ cm}^2 \text{ V}^{-1} \text{ S}^{-1}$ at RT (Cu _{1.97} S) σ : 80 S cm^{-1} at RT	Bulk Cu _{1.97} S pellets obtained by spark plasma sintering
[B2]	Cu _{2-x} S	n : $7.92 \times 10^{19} \text{ cm}^{-3}$ at RT (Cu ₂ S _{0.9} Se _{0.1}) μ : $6.02 \text{ cm}^2 \text{ V}^{-1} \text{ S}^{-1}$ at 300 K (Cu ₂ S _{0.9} Se _{0.1}) σ : 56.78 S cm^{-1} at RT	Bulk Cu ₂ S _{0.9} Se _{0.1} pellets obtained by spark plasma sintering
[B3]	Cu _{2-x} S	n : $\sim 10^{19} \text{ cm}^{-3}$ at RT μ : $4.28 \text{ cm}^2 \text{ V}^{-1} \text{ S}^{-1}$ (spin-coating) σ 75 S cm^{-1} at RT (EPD) 5.7 S cm^{-1} at RT (spin-coating)	Cu _{1.94-1.96} S nanoparticles thin films obtained by electrophoretic deposition (EPD) and spin-coating
[B4]	Cu _{2-x} S	n : N/A μ : N/A σ : 20.25 S cm^{-1} at RT	Pseudo-3D printed Cu _{1.97} S bulk
This work	Cu _{2-x} S	n : $6.29 \times 10^{19} \text{ cm}^{-3}$ at RT μ : $3.89 \text{ cm}^2 \text{ V}^{-1} \text{ s}^{-1}$ σ : 10 S cm^{-1} at RT	DLP-printed Cu _{2-x} S film
[B5]	SnSe ₂	n : $2.26 \times 10^{18} \text{ cm}^{-3}$ at RT μ $31.6 \text{ cm}^2 \text{ V}^{-1} \text{ S}^{-1}$ (ab plane) at RT $13.2 \text{ cm}^2 \text{ V}^{-1} \text{ S}^{-1}$ (c axis) at RT σ 11.4 S cm^{-1} (ab plane) at RT 4.8 S cm^{-1} (c axis) at RT	SnSe ₂ single crystals synthesised by a temperature gradient method.
[B6]	SnSe ₂	n : $8.4 \times 10^{17} \text{ cm}^{-3}$ at RT μ : N/A σ : $\sim 2 \text{ S cm}^{-1}$ at 350 K (cross plane)	Textured SnSe ₂ nanostructured bulk materials obtained by hot-pressing of SnSe ₂ nanoplates
[B7]	SnSe ₂	n : N/A μ : N/A σ : $\sim 10 \text{ S cm}^{-1}$ at RT	SnSe ₂ thin film fabricated by spin-coating of SnSe ₂ nanocrystals
This work	SnSe ₂	n : $\sim 10^{19} \text{ cm}^{-3}$ at RT μ : $2.25 \text{ cm}^2 \text{ V}^{-1} \text{ s}^{-1}$ σ : 2.79 S cm^{-1} at RT	DLP-printed SnSe ₂ film

Accordingly, we included the Table B1 and the following discussion in the revised manuscript

(page 12) and **Supplementary Information** (Supplementary Table 2).

“and are comparable with or even higher than those of the reported solution-processes thin films or the 3D-printed samples with the same materials (Supplementary Table 2). Although the electrical properties such as mobility and conductivity of our samples were roughly an order of magnitude lower than the bulk values, this can be understood with consideration that the ink-processed materials are usually more defective and less dense and include smaller grains compared with bulk materials synthesized by energy-intensive methods under harsh condition. For example, the grain sizes of our Cu₂S and SnSe₂ samples range from several tens to several hundreds of nanometres, which are several orders of magnitudes lower than those observed in the reported bulks^{52,54,55}. These smaller grains should cause the higher density of interfaces that can hinder the charge carrier transport. Moreover, various types of defects can be formed in these materials since the inks are the mixtures of organic solvents and precursors. In our samples, the precursors are organometallic complexes of ChaMs with counter cation of ethylenediammonium. Thus, there can be undesired impurities residues in the fabricated samples, which can reduce the charge carrier mobility and electrical conductivity by the impurity scattering. Finally, the ink-processed materials generally have porous features in the microstructures; e.g. the SEM images of our samples shows multiple pinholes in the nanometre scale. The formation of these pores is usually inevitable since they are created during the solvent drying or precursor decomposition that releases gases. All these reasons can be responsible to the reduced charge carrier mobility and resulting electrical conductivity.”

Comment 2: There are many printing methods available to print the inorganic inks. I suggest the authors provide a performance comparison of the optical printing vs. other printing methods in terms of printing speed, spatial resolution, properties of printed materials, etc.

Response: We appreciate the reviewer’s valuable comment. As the reviewer suggested, we summarized the recently reported printing methods for inorganic materials in terms of printable materials, printing speed, spatial resolution, processing steps, and the properties of materials (Table B2). **Our printing process are advantageous for the printing speed and simplicity in the printing process compared with other methods including inkjet printing, transfer printing, extrusion-based 3D printing, and other optical printing processes.** In addition, **the properties of the printed materials, especially electrical properties, are comparable to or even higher than the films printed by other process** except the direct optical lithography, which adapts the typical photolithographic techniques on the thin films for patterning. In addition, **our digital light processing (DLP)-based optical printing**, which creates solid architectures directly from liquid resin inks using automatically patterned digital masks and light exposure, **doesn’t need mask production, material deposition, and subsequent lift-off process**, thus which can additionally be **beneficial for reducing the processing cost and difficulty.**

Table B2. Comparison of the current optical printing method with the recent state-of-the-art printing methods for inorganics.

Printing method [ref]	Printable materials	Printing speed and resolution	Printing steps	Properties (ρ : resistivity R : resistance μ : carrier mobility σ : electrical conductivity S : Seebeck coefficient)	Remark
Direct patterning [B10]	Metal chalcogenides	Printing speed ~ 2 h (exposure) Resolution ~ 50 nm	Three steps (Coating, exposure and developing)	Sb ₂ S ₃ ρ : $2.7 \times 10^5 \Omega \text{ m}$ PbS μ : $1.8 \text{ cm}^2 \text{ V}^{-1} \text{ S}^{-1}$	Several lithographic technologies are applicable (Electron beam lithography, two-photon absorption lithography, thermal scanning probe lithography, UV light lithography)
Direct optical lithography [B11]	Metal, semiconductor, oxide nanocrystals	Printing speed Tens of second (exposure) Resolution ~ 1 μm	Three steps (Coating, exposure and developing)	Au ρ : $5.2 \times 10^{-8} \Omega \text{ m}$ CdSe μ : $20 \sim 100 \text{ cm}^2 \text{ V}^{-1} \text{ S}^{-1}$ IGZO μ : $4 \sim 10 \text{ cm}^2 \text{ V}^{-1} \text{ S}^{-1}$	Direct photolithography of spin-coated nanocrystal film
Light-induced material deposition (laser writing) [B12]	Metals and insulator (Iron oxide)	Printing speed Exposure time per pixel 30 ms ~ 1 s, Pixel size 446 nm ~ 1.17 μm Resolution 500 nm ~ 1 μm	Two steps (Exposure and cleaning)	Pt R : 12.8 Ω (18.8% of bulk) Ni R < 100 Ω Zn R > 10 k Ω Fe R > 200 M Ω Au R > 200 M Ω	Laser writing on the drop-casted metallate solution
Ink lithography (Ink-jet printing) [B13]	Metal, semiconductor, oxide, perovskite nanocrystals	Printing speed 300 mm s ⁻¹ Resolution ~ 35 μm	Three steps (Coating, printing, and developing)	Ag resistivity: $3.2 \pm 1.1 \times 10^{-5} \Omega \text{ m}$ PbSe μ : $1.0 \pm 0.3 \times 10^{-1} \text{ cm}^2 \text{ V}^{-1} \text{ S}^{-1}$ CdSe μ : $1.5 \pm 0.5 \text{ cm}^2 \text{ V}^{-1} \text{ S}^{-1}$	Inkjet printing, pen writing or brush painting are applicable Ligand inks printed on spin-coated nanocrystal film.
Ink-jet printing [B14]	Perovskite	Printing speed Thousands of seconds Resolution ~ 100 μm	Three steps (Coating, printing, and drying)	Photoluminescence quantum yield ~ 80%	Variation of pattern size by nozzle and drying temperature Printed on the polymer film
Transfer printing [B15]	Quantum dot nanocrystals	Printing speed Pick-up 10 cm s ⁻¹ Detach < 1 mm s ⁻¹ Resolution ~ 1 μm	Five steps (Coating, pick-up, contact, detach, and transfer)	Electroluminescence performance 14,000 cd m ⁻² at 7 V Quantum yield > 80%	Intaglio transfer printing. Need to fabricate a PDMS stamp and intaglio trench
Pseudo-3D printing [B6]	Cu _{2-x} S	Printing speed - Resolution Size of mould	Two steps (Moulding and pouring)	σ : 20.25 S cm ⁻¹ at RT S : ~100 $\mu\text{V K}^{-1}$ at 600 K	Need moulding apparatus

Extrusion-based 3D printing [B16]	Cu ₂ Se	Printing speed 3 mm s ⁻¹ Resolution ~ 330 μm	One step	σ : 183.42 S cm ⁻¹ S: 185.4 μV K ⁻¹	Topologically designed 3D printing for thermoelectric
Direct ink writing [B17]	p: Bi _{0.55} Sb _{1.45} Te ₃ n: Bi ₂ Sb _{2.7} Se _{0.3}	Printing speed 2 mm s ⁻¹ Resolution ~ 180 μm	One step	σ : 650 (p), 793.2 (n) S cm ⁻¹ S: 196.6 (p), -110.1(n) μV K ⁻¹	Direct-written filaments-based micro-TE devices
This work (DLP-based optical printing)	Metal chalcogenides (ChAM-based inks)	Printing speed Few ~ tens second (exposure) Resolution ~ 25 μm	Two steps (Printing and rinsing)	P-type Cu ₂ S μ : 3.89 cm ² V ⁻¹ s ⁻¹ σ : 10 S cm ⁻¹ at RT S: 87 μV K ⁻¹ at RT N-type SnSe ₂ μ : 2.25 cm ² V ⁻¹ s ⁻¹ σ : 2.79 S cm ⁻¹ at RT S: -203 μV K ⁻¹ at RT	Directly printing from inks Need no photomask or mould High-throughput printing at once Compatibility with 3D printing

Accordingly, we included the Table B2 (Supplementary Table 3) and the following related discussion in the revised manuscript (page 17).

“We further summarised the recently reported printing methods for inorganic materials in terms of printable materials, printing speed, spatial resolution, processing steps, and the properties of materials (Supplementary Table 3). Our printing process are advantageous for the printing speed and simplicity in the printing process compared with other methods including inkjet printing, transfer printing, extrusion-based 3D printing, and other optical printing processes.”

Comment 3: The authors demonstrated an in plane TE device. The optical printing method seems a very good method to fabricate cross-plane TE devices. The cross-plane TE devices are much more applicable for energy harvesting devices than in plane TE devices. If possible, the authors may attempt to demonstrate some cross-plane TE devices, which can make this work even more appealing.

Response: We appreciate the reviewer’s valuable suggestion. We have tried our best to fabricate the cross-plane thermoelectric device by the current DLP printing process. Fig. B1a shows the schematic illustration of the entire fabrication process, in which we used one pair of the DLP-printed Cu₂S and SnSe₂ films as p-type and n-type thermoelectric semiconductors. Since the current printing process is not applicable to electrode materials, we used the deposited Au and Ag paste layers as the bottom and top electrodes, respectively. The device resistance of ~150 Ω was observed at room temperature. We measured the power-generating performance of the fabricated generator by heating the bottom with

a hot plate and air-cooling the top of the device. Because the Cu_2S and SnSe_2 layers have an extremely thin thickness of ~ 500 nm, we couldn't obtain a temperature difference higher than 1 K in the steady-state. Also, the active cooling system couldn't be applied due to the relatively narrow length of less than 5 mm in the active part, including semiconductors and the top electrode. However, **the device showed an increase in the output voltages by two times and output powers by four times under the temperature difference of 0.2 and 0.4 K, respectively, demonstrating the reliable power generating performance** (Fig. B1b). We believe that this result clearly demonstrates the feasibility of our printing process to fabricate the 3D cross-plane thermoelectric device.

Fig. B1 (a) Scheme for the fabrication of the cross-plane thermoelectric generator consisting of the DLP-printed Cu_2S and SnSe_2 semiconductor layers (scale bar: 5 mm). **(b)** Output voltages and powers of the fabricated cross-plane thermoelectric generator.

Accordingly, we included Fig. B1 (Supplementary Fig. 20) and the following discussion in the revised manuscript (page 16).

“We further fabricate the cross-plane thermoelectric device by the DLP printing process. The schematic illustration (Supplementary Fig. 20a) shows the entire fabrication process, in which we used one pair of the DLP-printed Cu_2S and SnSe_2 films as p-type and n-type thermoelectric semiconductors. Since the current printing process is not applicable to electrode materials, we used the deposited Au and Ag paste layers as the bottom and top electrodes, respectively. The device resistance of $\sim 150 \Omega$ was observed at room temperature. We measured the power-generating performance of the fabricated generator by heating the bottom with a hot plate and air-cooling the top of the device. Because the Cu_2S and SnSe_2 layers have an extremely thin thickness of ~ 500 nm, we couldn't obtain a temperature difference higher than 1 K in the steady-state. However, the device showed an increase in the output voltages by two times and output powers by four times as the temperature difference increased from 0.2 K to 0.4 K, respectively, demonstrating the reliable power generating performance (Supplementary Fig. 20b). Such power generation performance observed in the in-plane and cross-plane clearly demonstrates the potential of our DLP-based optical printing process for fabricating integratable micro-scale 2D or 3D devices with diverse heterostructured functional materials. Indeed, our process may ultimately be able to facilitate the integration of electronic devices into other systems.”

References

- B1. He, Y. *et al.* High thermoelectric performance in non-toxic earth-abundant copper sulfide. *Adv. Mater.* **26**, 3974-3978 (2014).
- B2. Yao, Y., Zhang, B.-P., Pei, J., Liu, Y.-C & Li, J.-F. Thermoelectric performance enhancement of Cu₂S by Se doping leading to a simultaneous power factor increase and thermal conductivity reduction. *J. Mater. Chem. C* **5**, 7845-7852 (2017).
- B3. Otelaja, O. O., Ha, D. H., Ly, T., Zhang, H. & Robinson, R. D. Highly conductive Cu_{2-x}S nanoparticle films through room-temperature processing and an order of magnitude enhancement of conductivity via electrophoretic deposition. *ACS Appl. Mater. Interfaces* **6**, 18911-18920 (2014).
- B4. Burton, M. R., Mehraban, S., McGettrick, J., Watson, T., Lavery, N. P. & Carnie, M. J. Earth abundant, non-toxic, 3D printed Cu_{2-x}S with high thermoelectric figure of merit. *J. Mater. Chem. A* **7**, 25586-25592 (2019).
- B5. Pham, A.-T. *et al.* High-Quality SnSe₂ Single Crystals: Electronic and Thermoelectric Properties. *ACS Appl. Energy Mater.* **3**, 10787-10792 (2020).
- B6. Zhang, Y. *et al.* Tin Diselenide Molecular Precursor for Solution-Processable Thermoelectric Materials. *Angew. Chem.* **130**, 17309-17314 (2018).
- B7. Yin, D., Liu, Y., Dun, C., Carroll, D. L. & Swihart, M. T. Controllable colloidal synthesis of anisotropic tin dichalcogenide nanocrystals for thin film thermoelectrics. *Nanoscale* **10**, 2533-2541 (2018).
- B8. Li, F. *et al.* Ag-doped SnSe₂ as a promising mid-temperature thermoelectric material. *J. Mater. Sci.* **52**, 10506-10516 (2017).
- B9. Liang, X., Jin, D. & Dai, F. Phase Transition Engineering of Cu₂S to Widen the Temperature Window of Improved Thermoelectric Performance. *Adv. Electron. Mater.* **5**, 1900486 (2019).
- B10. Wang, W., Pfeiffer, P. & Schmidt-Mende, L. Direct Patterning of Metal Chalcogenide Semiconductor Materials. *Adv. Funct. Mater.* **30**, 2002685 (2020).
- B11. Wang, Y., Fedin, I., Zhang, H. & Talapin, D. V. Direct optical lithography of functional

- inorganic nanomaterials. *Science* **357**, 385-388 (2017).
- B12. Chen, Y. *et al.* A universal method for depositing patterned materials in situ. *Nat. Commun.* **11**, 5334 (2020).
- B13. Ahn, J. *et al.* Ink-Lithography for Property Engineering and Patterning of Nanocrystal Thin Films. *ACS Nano* **15**, 15667-15675 (2021).
- B14. Shi, L., *et al.* In Situ Inkjet Printing Strategy for Fabricating Perovskite Quantum Dot Patterns. *Adv. Funct. Mater.* **29**, 1903648 (2019).
- B15. Choi, M. K. *et al.* Wearable red-green-blue quantum dot light-emitting diode array using high-resolution intaglio transfer printing. *Nat. Commun.* **6**, 7149 (2015).
- B16. Choo, S. *et al.* Cu₂Se-based thermoelectric cellular architectures for efficient and durable power generation. *Nat. Commun.* **12**, 3550 (2021).
- B17. Kim, F. *et al.* Direct ink writing of three-dimensional thermoelectric microarchitectures. *Nat. Electron.* **4**, 579-587 (2021).

Reviewer #1 (Remarks to the Author):

The authors have replied in a satisfactory manner to most of my comments and I can appreciate that a great effort has been done to improve the quality of the manuscript. There are only a few small details that still need clarification, after this the paper is, in my opinion, ready for publication.

Comment#2: The authors add a good comparison but it fails to show the advantages of DLP vs DIW. If DIW is better in every aspect, why use DLP? Could the authors mention explicitly some advantages of DLP vs DIW? Maybe resolution or throughput?

"This is the stark contrast" should be -> "This is in stark contrast"

Comment #12: In their response, the authors claimed that "the hole carrier concentration is generally increased with increasing the copper deficiency from Cu_{2-x}S to Cu_{2-x}S ." I agree with this statement but this contradicts what is written in the text: "decreasing the number of Cu vacancy defects because of increasing the copper deficiency from Cu_{2-x}S to Cu_{2-x}S ". More Cu deficiency should lead to more vacancies, right? In this work, if S is evaporated, then the Cu content should be actually increasing (compared to S, i.e. Cu_{2-x}S becomes Cu_{2+x}S) and this will lead to reducing the vacancies and the carrier concentration. This will agree with the results but the explanation provided by the authors seems wrong then... Did not I understand it well?

Comment #15: The explanation is convincing, but the sentence "which shows the shift of the peak temperatures" is not clear. Please rephrase. Do you mean the shift of the peak value of the Seebeck coefficient to a different temperature?

In new table B2. Why inkjet printing consists of 3 steps? It should be printing and drying/annealing, as coating and patterning are done simultaneously.

Francisco Molina-Lopez

Reviewer #2 (Remarks to the Author):

The authors have satisfactorily addressed all the reviewer comments. I would like to recommend publication of this paper.

Response to the reviewers' comments

The followings are the responses to the reviewers' comments for the manuscript "Generalised optical printing of photocurable metal chalcogenides."

▪ **Reviewer #1**

General comment: The authors have replied in a satisfactory manner to most of my comments and I can appreciate that a great effort has been done to improve the quality of the manuscript. There are only a few small details that still need clarification, after this the paper is, in my opinion, ready for publication.

Response: We appreciate the reviewer's valuable time and effort in evaluating our manuscript. We truly agree with the reviewer's comments about the need for the additional characterisation and clarifications of the process and materials. We sincerely addressed all the comments and believe that these revisions made a significant improvement in the quality of the manuscript. We thank the reviewer again for the positive comment.

Comment 1: Comment#2: The authors add a good comparison but it fails to show the advantages of DLP vs DIW. If DIW is better in every aspect, why use DLP? Could the authors mention explicitly some advantages of DLP vs DIW? Maybe resolution or throughput?

"This is the stark contrast" should be -> "This is in stark contrast"

Response: We appreciate the reviewer's valuable comment. Optical printing methods such as DLP and SLA can be advantageous for high-resolution, high-throughput, and large-scale printing, compared with other printing methods, especially the DIW.

Accordingly, we included the following sentences in the revised manuscript (page 3). Also, we have tried to edit all grammatical errors in the entire manuscript.

"Nevertheless, the optical printing methods of DLP and SLA can be advantageous for high-resolution, high-throughput, and large-scale printing, which can offer great potential for patterning high-performance inorganic materials."

Comment 2: Comment #12: In their response, the authors claimed that "the hole carrier concentration is generally increased with increasing the copper deficiency from Cu_{2-x}X to Cu_{2-x}X ." I agree with this statement but this contradicts what is written in the text: "decreasing the number of Cu vacancy defects because of increasing the copper deficiency from Cu_{2-x}X to Cu_{2-x}X ". More Cu deficiency should lead to more vacancies, right? In this work, if S is evaporated, then the Cu content should be actually increasing (compared to S, i.e. Cu_{2-x}X becomes Cu_{2+x}X) and this will lead to reducing the vacancies and the carrier concentration. This will agree with the results but the explanation provided by the authors seems wrong then... Did not I understand it well?

Response: We apologize for the confusing description in the previous manuscript. As the reviewer commented, the S evaporation lead to the increase of the Cu content, reducing the Cu vacancies and hole concentrations.

Accordingly, we replaced the previous sentences with the following sentences in the revised manuscript (page 11).

"Because the intrinsic defects of Cu vacancies in Cu_2S act as hole donors, longer annealing times could promote the evaporation of S, eventually decreasing the hole concentrations by increasing the relative Cu contents and reducing the Cu vacancy defects⁵³."

Comment 3: Comment #15: The explanation is convincing, but the sentence "which shows the shift of the peak temperatures" is not clear. Please rephrase. Do you mean the shift of the peak value of the Seebeck coefficient to a different temperature?

Response: We apologize for the unclear sentences in the revised manuscript. As the reviewer noted, we intended to describe that the peak value of the Seebeck coefficient could shift to higher temperature ranges with increasing the carrier concentration.

Accordingly, we replaced the previous sentences with the following sentences in the revised manuscript (page 14).

"Accordingly, these materials usually show the positive temperature dependences in a wide temperature range. This behaviour is generally reflected in the temperature-dependent Seebeck coefficients of materials, which shows

the shift of the peak value of the Seebeck coefficients to higher temperature ranges with the increase of carrier concentrations of materials^{61,62.}”

Comment 4: In new table B2. Why inkjet printing consists of 3 steps? It should be printing and drying/annealing, as coating and patterning are done simultaneously.

Response: We appreciate the reviewer’s thoughtful comment. Generally, the inkjet printing method consists of two step processes of printing and drying/ annealing. However, the references that we cited in the manuscript utilized three-step or four-step processes for securing high-performance of nanocrystal or perovskite films (Fig. A1).^{A1,A2}. For example, in the first paper (Ahn, J. et al. Ink-Lithography for Property Engineering and Patterning of Nanocrystal Thin Films. *ACS Nano* **15**, 15667-15675 (2021))^{A1} the entire printing process consists of three steps of i) coating the nanocrystal films, ii) inkjet printing of ligand inks, and iii) stripping unexposed region of nanocrystal films (Fig. A1a). In the second paper (Shi, L. et al. In Situ Inkjet Printing Strategy for Fabricating Perovskite Quantum Dot Patterns. *Adv. Funct. Mater.* **29**, 1903648 (2019))^{A2}, the printing processes for perovskite quantum dots is composed of i) dropping of polymer film, ii) inkjet printing of perovskite precursor inks, and iii) dissolving and drying (Fig. A1b). Accordingly, we described that the inkjet printing methods in the Supplementary Table 3 consist of three steps such as coating, printing, developing, or drying.

Fig. A1 (a) Scheme of the ink-lithography technique, in which i) spin-coated nanocrystal inks, ii) a ligand ink is inkjet-printed on the nanocrystal thin film, iii) unexposed regions are stripped (rinsing) (adapted from Ref. A1). (b) Scheme of the in situ inkjet printing strategy for perovskite quantum dots (adapted from Ref. A2).

References

- A1. Ahn, J. *et al.* Ink-Lithography for Property Engineering and Patterning of Nanocrystal Thin Films. *ACS Nano* **15**, 15667-15675 (2021).
- A2. Shi, L. *et al.* In Situ Inkjet Printing Strategy for Fabricating Perovskite Quantum Dot Patterns. *Adv. Funct. Mater.* **29**, 1903648 (2019).

▪ **Reviewer #2**

General comment: The authors have satisfactorily addressed all the reviewer comments. I would like to recommend publication of this paper.

Response: We appreciate the reviewer's valuable time and effort in evaluating our manuscript. We thank the reviewer again for the positive comment.

Reviewer #1 (Remarks to the Author):

The authors have addressed all the comments from the second round of the review in a satisfactory manner. I recommend the publication of this paper.

Francisco Molina-Lopez